# Changing the incentive structure of social media platforms to halt the spread of misinformation

Laura K Globig[1,2,3]*, Nora Holtz[1,2], Tali Sharot[1,2,3]*

[1]Affective Brain Lab, Department of Experimental Psychology, University College London, London, United Kingdom; [2]The Max Planck UCL Centre for Computational Psychiatry and Ageing Research, University College London, London, United Kingdom; [3]Department of Brain and Cognitive Sciences, Massachusetts Institute of Technology, Cambridge, United States

**Abstract** The powerful allure of social media platforms has been attributed to the human need for social rewards. Here, we demonstrate that the spread of misinformation on such platforms is facilitated by existing social 'carrots' (e.g., 'likes') and 'sticks' (e.g., 'dislikes') that are dissociated from the veracity of the information shared. Testing 951 participants over six experiments, we show that a slight change to the incentive structure of social media platforms, such that social rewards and punishments are contingent on information veracity, produces a considerable increase in the discernment of shared information. Namely, an increase in the proportion of true information shared relative to the proportion of false information shared. Computational modeling (i.e., drift-diffusion models) revealed the underlying mechanism of this effect is associated with an increase in the weight participants assign to evidence consistent with discerning behavior. The results offer evidence for an intervention that could be adopted to reduce misinformation spread, which in turn could reduce violence, vaccine hesitancy and political polarization, without reducing engagement.

*For correspondence:
laura.globig@gmail.com (LKG);
t.sharot@ucl.ac.uk (TS)

Competing interest: The authors declare that no competing interests exist.

## Editor's evaluation

This important paper outlines a novel method for reducing the spread of misinformation on social media platforms. A compelling series of experiments and replications support the main claims, which could have significant real-world societal impact.

## Introduction

In recent years, the spread of misinformation online has skyrocketed, increasing polarization, racism and resistance to climate action and vaccines (*Barreto et al., 2021*; *Rapp and Salovich, 2018*; *Tsfati et al., 2020*; *Van Bavel et al., 2021*). Existing measures to halt the spread, such as flagging posts, have had limited impact (e.g., *Chan et al., 2017*; *Grady et al., 2021*; *Lees et al., 2022*).

We hypothesize that the spread of misinformation on social media platforms is facilitated by the existing incentive structure of those platforms, where social rewards (in the form of 'likes' and 'shares') are dissociated from the veracity of the information (*Figure 1a*, left panel, *Sharot, 2021*). The rationale for this hypothesis is as follows: users can discern true from false content to a reasonable degree (*Allen et al., 2021*; *Pennycook and Rand, 2019*). Yet, because misinformation generates no less retweets and 'likes' than reliable information (*Lazer et al., 2018*; *Vosoughi et al., 2018*), and online behavior conforms to a reinforcement-learning model by which users are reacting to social rewards (*Lindström et al., 2021*; *Brady et al., 2021*) users have little reason to use their discernment to guide

**eLife digest** In recent years, the amount of untrue information, or 'misinformation', shared online has increased rapidly. This can have profound effects on society and has been linked to violence, political extremism, and resistance to climate action.

One reason for the spread of misinformation is the lack of incentives for users to share true content and avoid sharing false content. People tend to select actions that they believe will lead to positive feedback ('carrots') and try to avoid actions that lead to negative feedback ('sticks'). On most social media sites, these carrots and sticks come in the form of 'like' and 'dislike' reactions, respectively. Stories that users think will attract 'likes' are most likely to be shared with other users. However, because the number of likes a post receives is not representative of how accurate it is, users share information even if they suspect it may not be accurate. As a result, misinformation can spread rapidly.

Measures aimed at slowing the spread of misinformation have been introduced to some social media sites, such as removing a few virulent spreaders of falsities and flagging misleading content. However, measures that change the incentive structure of sites so that positive and negative feedback is based on the trustworthiness of the information have not yet been explored.

To test this approach, Globig et al. set up a simulated social media site that included 'trust' and 'distrust' buttons, as well as the usual 'like' and 'dislike' options. The site featured up to one hundred news stories, half of which were untrue. More than 900 participants viewed the news posts and could react using the new buttons as well as repost the stories.

The experiment showed that participants used the 'trust' and 'distrust' buttons to differentiate between true and false posts more than the other options. As a result, to receive more 'trust' responses and less 'distrust' responses from other users, participants were more likely to repost true stories than false ones. This led to a large reduction in the amount of misinformation being spread. Computational modeling revealed that the participants were paying more attention to how reliable a news story appeared to be when deciding whether to repost it.

Globig et al. showed that adding buttons to highlight the trustworthiness of posts on social media sites reduces the spread of misinformation, without reducing user engagement. This measure could be easily incorporated into existing social media sites and could have a positive impact on issues that are often fuelled by misinformation, such as vaccine hesitancy and resistance to climate action.

their sharing behavior. Thus, people will share misinformation even when they do not trust it (*Pennycook et al., 2021*; *Ren et al., 2021*).

To halt the spread, an incentive structure is needed where 'carrots' and 'sticks' are directly associated with accuracy (*Figure 1a*, right panel, *Sharot, 2021*). Such a system will work with the natural human tendency to select actions that lead to the greatest reward and avoid those that lead to punishment (*Skinner, 1966*). Scientists have tested different strategies to reduce the spread of misinformation, including educating people about fake news (*Guess et al., 2020*; *Traberg et al., 2022*), using a prompt to direct attention to accuracy (*Kozyreva et al., 2020*; *Pennycook et al., 2021*; *Pennycook et al., 2020*) and limiting how widely a post can be shared (*Jackson et al., 2022*). Surprisingly, possible interventions in which the incentive structure of social media platforms is altered to reduce misinformation have been overlooked.

Here, we test the efficacy of such a structure by slightly altering the engagement options offered to users. Specifically, we add an option to react to posts using 'trust' and 'distrust' buttons (*Figure 1b*). We selected these buttons because trust by definition is related to veracity – it is defined as 'a firm belief in the reliability, truth, ability, or strength of someone or something' (Oxford Dictionary).

We hypothesize that (1) people will use the 'trust' and 'distrust' buttons to discern true from misinformation more so than the commonly existing engagement options (such as a 'like' button; *Figure 1b*, top panel). By 'discernment' we mean that true posts will receive more 'trusts' reactions than 'distrusts' reactions and vice versa for false posts. This will create an environment in which rewards ('trusts') and punishments ('distrusts') are more directly associated with the veracity of information. Thus, (2) when exposed to this environment, users will start sharing more true information and less false information in order to obtain more 'trust' carrots and fewer 'distrust' sticks (*Figure 1b*, bottom panel). The new

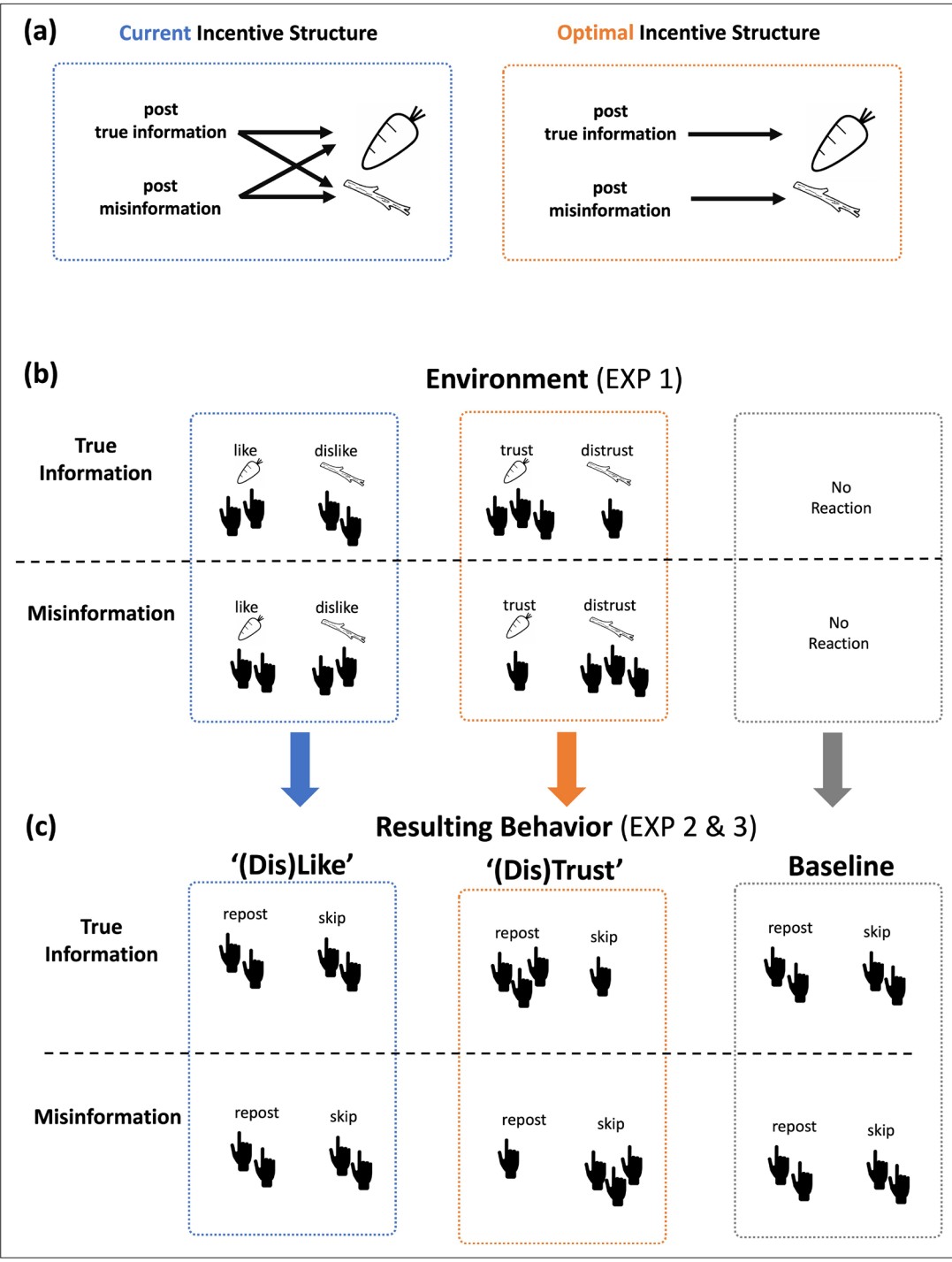

**Figure 1.** Theoretical framework. (**a**) The current incentive structure (blue) is such that the veracity of shared information is dissociated from rewards ('carrots') and punishments ('sticks'). That is, true information and misinformation may lead to roughly equal number of rewards and punishments. An optimal incentive structure (orange) is such that sharing true information is rewarded with more 'carrots' than sharing misinformation, which in turn is penalized with more 'sticks' than true information. To create an optimal environment, an intervention is needed by which the number of rewards and punishments are directly associated with the veracity of information. (**b**) We test one such possible intervention (Experiment 1). In particular, we allow people to engage with posts using 'trust' reaction buttons and 'distrust' reaction buttons (orange). The rationale is that they will use these reactions to discern true from false information more so than 'like' and 'dislike' reaction buttons. (**c**) As a result, to obtain a greater number of 'trust' carrots and a smaller number of 'distrust' sticks in response to a post, people in

*Figure 1 continued on next page*

*Figure 1 continued*
the optimal environment (orange) will share more true than misinformation compared to those in the suboptimal environment which includes no feedback at all (gray), and those in an environment where the association between veracity of information and number of carrots and sticks is weak (blue). This second step is tested in Experiments 2 and 3.

feedback options could both reinforce user behavior that generates trustworthy material and signal to others that the post is dependable.

We also test environments in which participants receive only 'trusts' (a different number of *trust* for different posts) or only 'distrusts' (a different number of *distrust* for different posts) to examine if and how the impact of small vs large positive feedback ('trust') on discernment differs from the impact of small vs large negative feedback (distrust'). It has been proposed that the possibility of reward is more likely to reinforce action than the possibility of punishment, while the possibility of punishment is more likely to reinforce inaction (**Guitart-Masip et al., 2014**; **Guitart-Masip et al., 2011**; **Guitart-Masip et al., 2012**). This may translate to a large number of 'trust' selectively increasing sharing of true information without decreasing sharing of misinformation and vice versa for large number of 'distrust'. Further, being mindful of potential differences in sharing behavior across political parties (e.g., **Grinberg et al., 2019**; **Guess et al., 2020**) we test participants from both sides of the political divide.

To that end, over six experiments 951 participants engaged in simulated social media platforms where they encountered true and false information. In Experiment 1, we examined whether participants would use 'trust' and 'distrust' buttons to discern true from false information more so than existing 'like' and 'dislike' buttons (**Figure 1b**, replication: Experiment 4). In Experiments 2 and 3, we tested whether new groups of participants would share more true than false information in social media platforms that introduce real 'trust' and 'distrust' feedback from other participants (**Figure 1c**, replication: Experiments 5 and 6). The intuition is that 'trust' and 'distrust' reactions will naturally be used to indicate veracity and thus provide a reward structure contingent on accuracy, thereby reducing the sharing of misinformation and generating a healthier information ecosystem. Using computational modeling we provide insights into the specific mechanism by which our intervention improves sharing discernment.

## Results

### Participants use 'trust' and 'distrust' buttons to discern true from false information (Experiment 1)

In a first step, we examined whether participants used 'trust' and 'distrust' reactions to discern true from false information more so than 'like' and 'dislike' reactions. In Experiment 1, participants saw 100 news posts taken from the fact-checking website Politifact (https://www.politifact.com; see *Figure 2*). Half of the posts were true, and half were false. Participants were given the opportunity

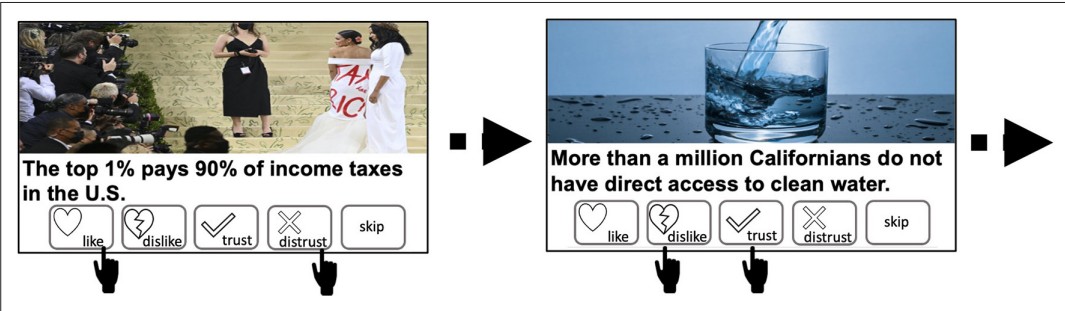

**Figure 2.** Task (Experiment 1). Participants observed a series of 100 posts in random order (50 true, 50 false). Their task was to react using one or more of the 'like', 'dislike', 'trust', or 'distrust' buttons or to skip. The task was self-paced.

The online version of this article includes the following figure supplement(s) for figure 2:

**Figure supplement 1.** Instructions for Experiment 1.

to react to each post using 'like', 'dislike', 'trust', and 'distrust' reaction buttons. They could select as many buttons as they wished or none at all (skip). Five participants were excluded according to pre-determined criteria (see Materials and methods for details). Thus, 106 participants (52 Democrats, 54 Republican, $M_{age}$ = 40.745, $SD_{age}$ ± 14.479; female = 54, male = 52) were included in the analysis. See *Figure 2—figure supplement 1* for full instructions.

We then examined whether participants used the different reaction buttons to discern true from false information. Discernment was calculated as follows, such that high numbers always indicate better discernment:

For 'like':

$$Discernment = Prop_{likes\ true} - Prop_{likes\ false}$$

For 'dislike':

$$Discernment = Prop_{dislikes\ false} - Prop_{dislikes\ true}$$

For 'trust':

$$Discernment = Prop_{trusts\ true} - Prop_{trusts\ false}$$

For 'distrust':

$$Discernment = Prop_{distrusts\ false} - Prop_{distrusts\ true}$$

With Prop indicating the proportion of that response out of all true posts, or out of all false posts, as indicated.

These discernment scores were calculated for each participant separately and then entered into a 2 (*type of reaction*: 'trust' and 'distrust'/'like' and 'dislike') by 2 (*valence of reaction*: positive, i.e., 'like', 'trust'/negative, i.e., 'dislike', 'distrust') within-subject analysis of variance (ANOVA). Political orientation was also added as a between-subject factor (Republican/Democrat), allowing for an interaction of political orientation and type of reaction to assess whether participants with differing political beliefs used the reaction buttons in different ways.

The results reveal that participants' use of '(Dis)Trust' reaction buttons (*M* = 0.127; SE = 0.007) was more discerning than their use of '(Dis)Like' reaction buttons (*M* = 0.047; SE = 0.005; $F_{(1,104)}$ = 95.832, p < 0.001, partial $\eta^2$ = 0.48, **Figure 3**). We additionally observed an effect of valence ($F_{(1,105)}$ = 17.33, p < 0.001, partial $\eta^2$ = 0.14), with negatively valenced reaction buttons (e.g., 'dislike' and 'distrust', *M* = 0.095, SE = 0.007) being used in a more discerning manner than positively valenced reaction buttons (e.g., 'like' and 'trust', *M* = 0.087, SE = 0.005) and an effect of political orientation ($F_{(1,104)}$ = 25.262, p < 0.001, partial $\eta^2$ = 0.2), with Democrats (*M* = 0.115, SE = 0.007) being more discerning than Republicans (*M* = 0.06, SE = 0.005). There was also an interaction of type of reaction and political orientation ($F_{(1,104)}$ = 24.084, p < 0.001, partial $\eta^2$ = 0.19), which was characterized by Democrats showing greater discernment than Republicans in their use of '(Dis)Trust' reaction buttons ($F_{(1,104)}$ = 33.592, p < 0.001, partial $\eta^2$ = 0.24), but not in their use of '(Dis)Like' reaction buttons ($F_{(1,104)}$ = 2.255, p = 0.136, partial $\eta^2$ = 0.02). Importantly, however, both Democrats ($F_{(1,51)}$ = 93.376, p < 0.001, partial $\eta^2$ = 0.65) and Republicans ($F_{(1,53)}$ = 14.715, p < 0.001, partial $\eta^2$ = 0.22) used the '(Dis)Trust' reaction buttons in a more discerning manner than the '(Dis)Like' reaction buttons.

One-sample *t*-tests against zero further revealed that participants' use of each reaction button discerned true from false information ('like': *M* = 0.06; SE = 0.006; $t_{(105)}$ = 10.483, p < 0.001, Cohen's *d* = 1.018; 'trust': *M* = 0.099; SE = 0.01; $t_{(105)}$ = 9.744, p < 0.001, Cohen's *d* = 0.946; 'dislike': *M* = 0.034; SE = 0.007; $t_{(105)}$ = 4.76, p < 0.001, Cohen's *d* = 0.462; 'distrust': *M* = 0.156; SE = 0.01; $t_{(105)}$ = 15.872, p < 0.001, Cohen's *d* = 1.542).

Thus far, we have shown that participants use '(Dis)Trust' reaction buttons in a more discerning manner than '(Dis)Like' reaction buttons. As social media platforms care about overall engagement not only its quality, we examined how frequently participants used the different reaction buttons. An ANOVA with the same specifications as above was conducted, but this time submitting frequency of reaction as the dependent variable. We found that participants used '(Dis)Trust' reaction buttons more often than '(Dis)Like' reaction buttons (percentage use of reaction out of all trials: 'trust': *M* = 28.057%; 'distrust': *M* = 34.085%; 'like': *M* = 18.604%; 'dislike': *M* = 23.745%; $F_{(1,104)}$ = 36.672, p <

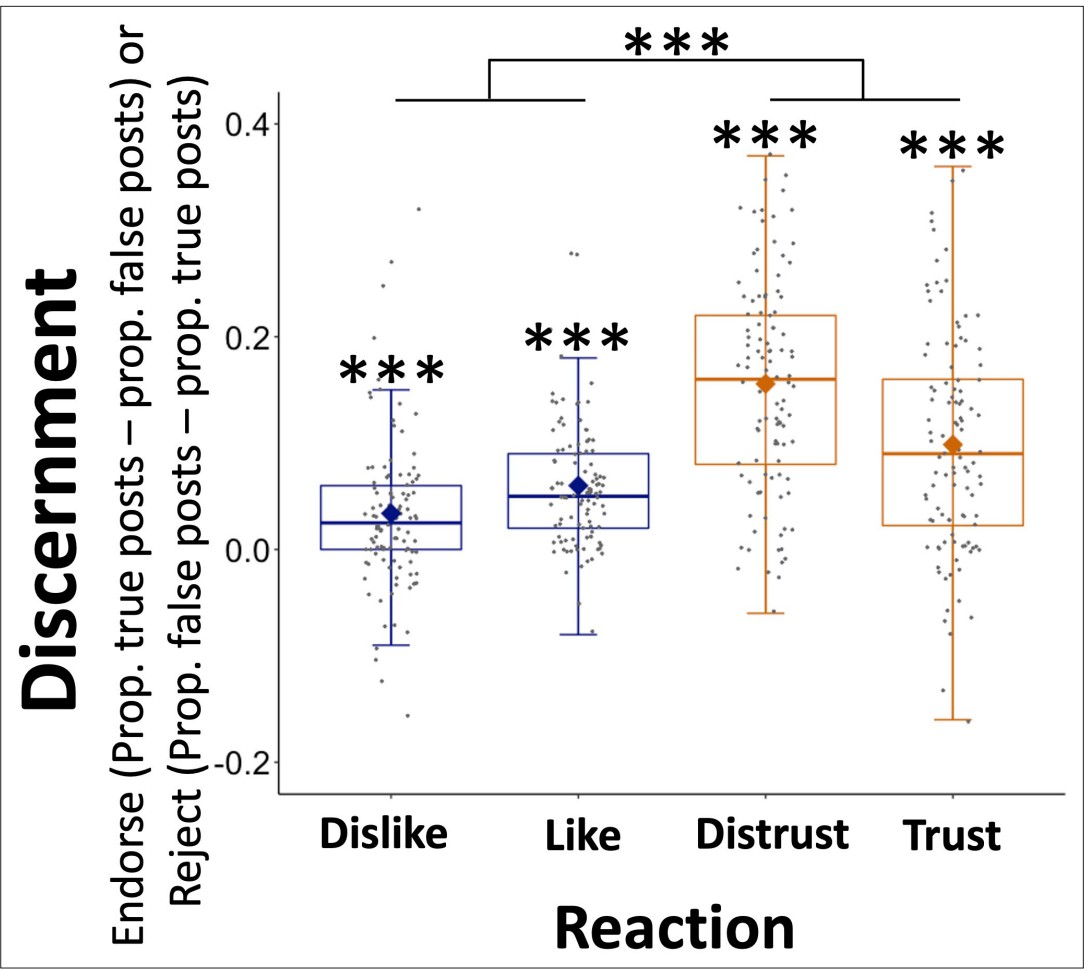

**Figure 3.** Participants use 'trust' and 'distrust' reactions to discern true from false information. 'Distrust' and 'trust' reactions were used in a more discerning manner than 'like' and 'dislike' reactions. *Y* axis shows discernment between true and false posts. For positive reactions (e.g., '*likes*' and '*trusts*'), discernment is equal to the proportion of positive reactions for true information minus false information, and vice versa for negative reactions ('*dislikes*' and '*distrusts*'). *X* axis shows reaction options. Data are plotted as box plots for each reaction button, in which horizontal lines indicate median values, boxes indicate 25/75% interquartile range and whiskers indicate 1.5 × interquartile range. Diamond shape indicates the mean discernment per reaction. Individuals' mean discernment data are shown separately as gray dots. Symbols above each box plot indicate significance level compared to 0 using a t-test. N=106, ***p < 0.001.

The online version of this article includes the following figure supplement(s) for figure 3:

**Figure supplement 1.** Participants' use '(Dis)Trust' buttons to discern true from false information (Experiment 4).

0.001, partial $\eta^2$ = 0.26). In addition, negative reaction buttons ('distrust' and 'dislike': *M* = 28.915%, SE = 1.177) were used more frequently than positive reaction buttons ('trust' and 'like': *M* = 23.33%, SE = 1.133; $F(1,105)$ = 16.96, p < 0.001, partial $\eta^2$ = 0.07). No other effect was significant. Interestingly, we also found that participants who skipped more posts were less discerning ($R$ = −0.414, p < 0.001). Together, the results show that the new reaction options increase engagement.

The results hold when controlling for demographics, when not including political orientation in the analysis, and allowing for an interaction between type of reaction and valence (see *Supplementary files 1 and 2*). The results also replicate in an independent sample (Experiment 4, see Materials and methods for details; and *Figure 3—figure supplement 1*, *Supplementary file 3*).

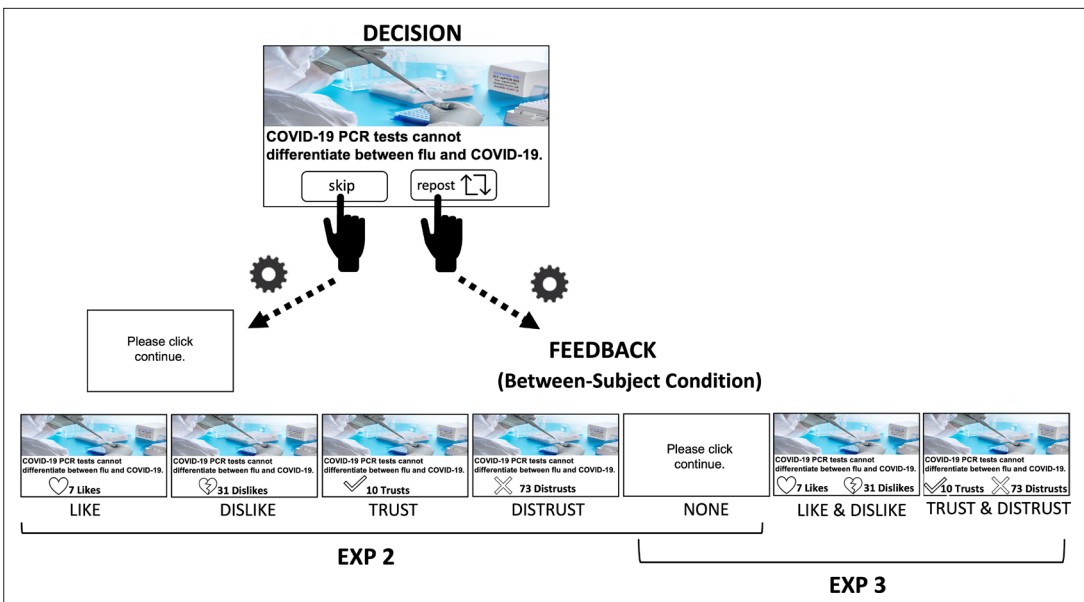

**Figure 4.** Task. In Experiment 2 on each of 100 trials participants observed a post (50 true, 50 false content). They then chose whether to share it or skip (self-paced). They were told that if they chose to share a post, it would be shared to their feed such that other participants would be able to see the post and react to it in real time (*feedback*). Depending on the environment participants were in, they could either observe the number of (1) '*dislikes*' (N = 45), (2) '*likes*' (N = 89), (3) '*distrusts*' (N = 49), or (4) '*trusts*' (N = 46) feedback. The feedback was in fact the number of reactions gathered from Experiment 1, though the participants believed the reactions were in real time as indicated by a rotating cogwheel (1 s). Once the feedback appeared, participants could then click continue. If participants selected to skip, they would observe a white screen asking them to click continue (self-paced). In the *Baseline* environment (N = 59) participants received no feedback. Experiment 3 was identical to Experiment 2 with two distinctions: (1) Depending on the environment participants were in, they could either observe the number of (i) both '*dislikes*' and '*likes*' (N = 128), (ii) both '*distrusts*' and '*trusts*' (N = 137), or (iii) no feedback (Baseline, N = 126). (2) In Experiment 3, we selected 40 posts (20 true, 20 false) to which Republicans and Democrats had on average reacted to similarly using the 'trust' button in Experiment 1. Discernment was calculated for each participant by subtracting the proportion of sharing false information from the proportion of sharing true information. High discernment indicates greater sharing of true than false information.

The online version of this article includes the following figure supplement(s) for figure 4:

**Figure supplement 1.** Instructions for Experiment 2.

**Figure supplement 2.** Instructions for Experiment 3.

## 'Trust' and 'distrust' incentives improve discernment in sharing behavior (Experiment 2)

Thus far, we have shown that participants use '(Dis)Trust' reaction buttons in a more discerning manner than '(Dis)Like' reaction buttons. Thus, an environment which offers '(Dis)Trust' feedback is one where the number of 'carrots' (in the form of 'trusts') and the number of 'sticks' (in the form of 'distrusts') are directly associated with the veracity of the posts. It then follows that submitting participants to such an environment will increase their sharing of true information (to receive 'trusts') and reduce their sharing of misinformation (to avoid 'distrusts').

To test this, we ran a second experiment. A new group of participants (N = 320) were recruited to engage in a simulated social media platform. They observed the same 100 posts (50 true, 50 false) shown to the participants in Experiment 1, but this time instead of reacting to the posts they could either share the post or skip it (see *Figure 4* and *Figure 4—figure supplements 1 and 2* for full instructions). They were told that if they chose to share a post, it would be shared to their feed such that the other participants would be able to see the post and would then be able to react to it in real time (*feedback*). Depending on the environment participants were in, which varied between subjects, they could receive feedback in the form of the number of users who (1) '*disliked*', (2) '*liked*', (3) '*distrusted*', or (4) '*trusted*' their posts. We also included a (5) baseline condition, in which participants

received no feedback. If participants selected to skip, they would observe a white screen asking them to click continue. Data of 32 participants were not analyzed according to pre-determined criteria (see Materials and methods for details). Two-hundred and eighty-eight participants (146 Democrats, 142 Republicans, $M_{age}$ = 38.073, $SD_{age}$ ± 13.683; female = 147, male = 141) were included in the analysis (see Materials and methods for details).

$$Discernment = Prop_{reposts\ true} - Prop_{reposts\ false}$$

These scores were submitted into a between-subject ANOVA with *type of feedback* ('trust' and 'distrust'/'like' and 'dislike'/Baseline), *valence* (positive, i.e., 'like', 'trust'/negative, i.e., 'dislike', 'distrust' vs neutral/no feedback) and political orientation (Republican/Democrat) as factors. We also allowed for an interaction of political orientation and type of feedback.

We observed an effect of type of feedback ($F(1,281)$ = 15.2, p < 0.001, partial $\eta^2$ = 0.051), such that participants shared more true than false information in the '(Dis)Trust' environments ($M$ = 0.18, SE = 0.018) than the '(Dis)Like' environments ($M$ = 0.085, SE = 0.019, $F(1,225)$ = 14.249, p < 0.001, partial $\eta^2$ = 0.06) and Baseline environment ($M$ = 0.084, SE = 0.025; $F(1,150)$ = 10.906, p = 0.001, partial $\eta^2$ = 0.068, *Figure 5a*). Moreover, participants who received 'trust' feedback ($M$ = 0.176, SE = 0.026) were more discerning in their sharing behavior than those who received 'like' feedback ($M$ = 0.081, SE = 0.021, $F(1,131)$ = 10.084, p = 0.002, partial $\eta^2$ = 0.071). Those who received 'distrust' feedback ($M$ = 0.175, SE = 0.026) were more discerning than those who received 'dislike' feedback ($M$ = 0.092, SE = 0.039, $F(1,90)$ = 5.003, p = 0.028, partial $\eta^2$ = 0.053). We further observed a trend interaction between type of feedback and political orientation ($F(1,281)$ = 2.939, p = 0.055, partial $\eta^2$ = 0.02). While Democrats ($M$ = 0.213; SE = 0.014) were generally more discerning than Republicans ($M$ = 0.017; SE = 0.016; $F(1,281)$ = 77.392, p < 0.001, partial $\eta^2$ = 0.216), this difference was smaller in those who received '(Dis)Trust' feedback ($M$ = 0.082, SE = 0.034) compared to those who received '(Dis)Like' feedback ($M$ = 0.23, SE = 0.03; $F(1,224)$ = 4.879, p = 0.028, partial $\eta^2$ = 0.021) and by trend smaller than those who received no feedback ($M$ = 0.229, SE = 0.045; $F(1,149)$ = 3.774, p = 0.054, partial $\eta^2$ = 0.025). There was no difference between the latter two ($F(1,188)$ = 0.00, p = 0.988, partial $\eta^2$ = 0.00). No other effects were significant. Overall engagement, measured as percentage of posts shared out of all trials, did not differ across environments ($F(1,281)$ = 1.218, p = 0.271, partial $\eta^2$=0.004; Mean % posts shared out of all trials: Baseline = 27.712%; Dislike = 35.889%; Like = 33.258%; Distrust = 32.51%; Trust = 30.435%; see *Supplementary file 4* for means for true and false posts).

Results hold when controlling for demographics, when not including political orientation in the analysis, and allowing for an interaction between type of reaction and valence (see *Supplementary files 5 and 6*). Results replicate in an independent sample (Experiment 5, see Materials and methods for details; and *Figure 5—figure supplement 1*, *Supplementary file 7*).

To recap – participants in Experiment 2 decided whether to share content or skip. They then observed the reaction of other participants to their post (they believed this was happening in real time, but for simplicity we fed them reactions of participants from Experiment 1). Each participant in Experiment 2 observed only one type of feedback. For example, only 'distrusts'. How is it that observing 'distrusts' alone increases discernment? The rationale behind this design is that for any given post, true or false, some users will distrust the post. However, true posts will receive fewer 'distrusts' than false posts. It is the number of 'distrusts' per post that matters. The participants are motivated to minimize the average number of 'distrusts' they receive. To achieve this, they should post more true posts and fewer false posts. Of course, if the participants were simply trying of minimize the *total* number of distrusts, they would just skip on every trial. Participants do not do that, however. Potentially because sharing in and of itself is rewarding (*Tamir and Mitchell, 2012*). The results indicate that participants are sensitive to the number of 'distrusts' per posts not just to the total number of 'distrusts' over all posts.

The same rationale holds for the participants that only observe 'trusts'. They receive more 'trusts' for true than false posts. It is the magnitude of 'trusts' that is associated with veracity. This motivates participants to post more true posts and fewer false posts in order to maximize the average number of 'trusts' per post. Of course, if participants were simply trying of maximize the total number of 'trusts', they would just share on every trial. Participants do not do that, however. This indicates that they are sensitive to the number of 'trusts' per post not just to total number over all posts. Any user of social media platforms could relate to this; when posting a tweet, for example, many people will

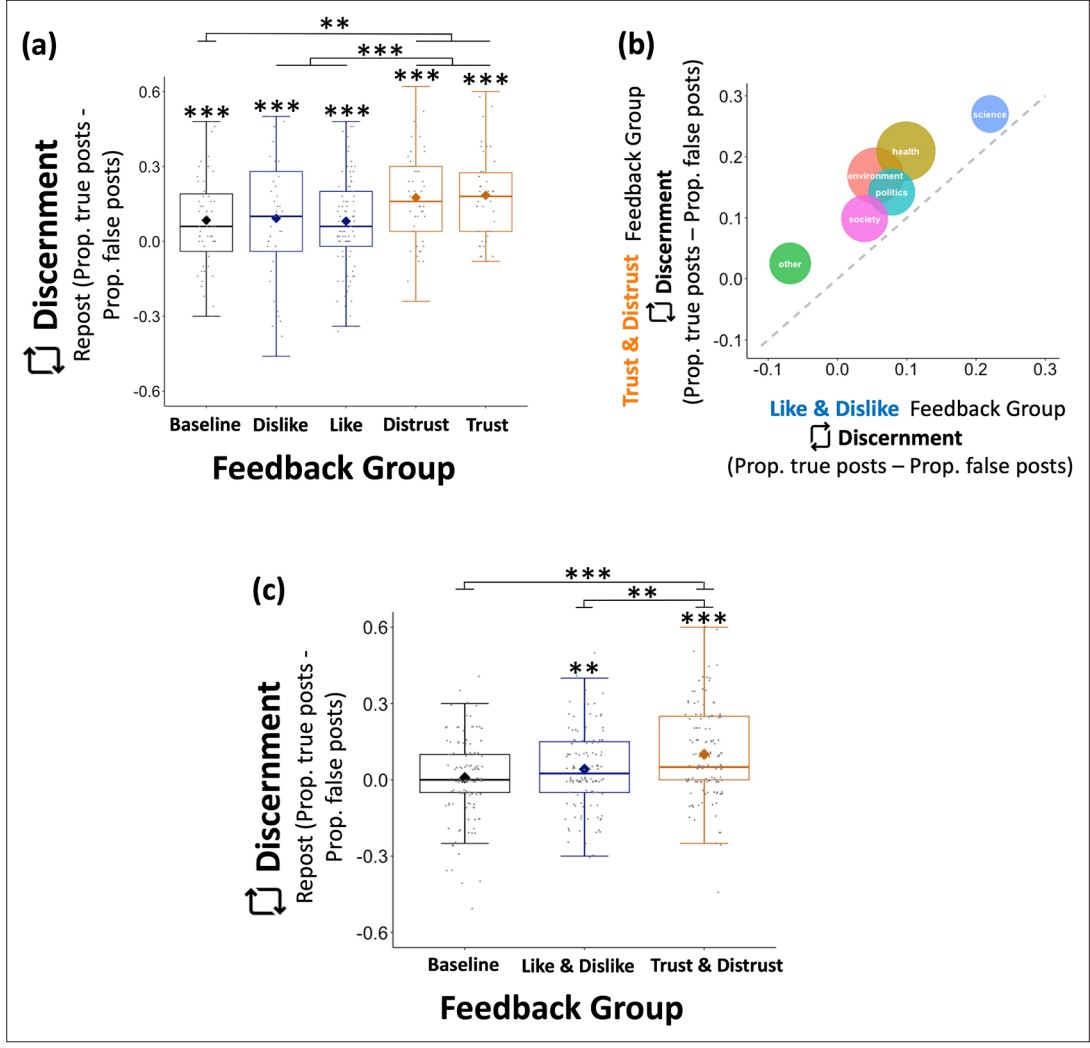

**Figure 5.** Altering the incentive structure of social media environments increases discernment of information shared. (**a**) Participants (N=288) operating in an environment where '(Dis)Trust' feedback was introduced shared more true information relative to false information than participants operating in an environment where only '(Dis) Like' feedback was available, or no feedback at all (Baseline) *Y* axis shows discernment, that is, proportion of true posts shared minus proportion of false posts shared. *X* axis shows the group environment (type of feedback). (**b**) This was the case regardless of the topic of the post (politics, science, health, environment, society, other). Bubble size corresponds to number of the posts included in the study. Diagonal dashed line indicates point of equivalence, where discernment in equal across the '(Dis)Like' and '(Dis)Trust' environments. As can be seen, all circles are above the dashed line indicating that in all cases discernment is greater in an environment that offers '(Dis)Trust' feedback. *Y* axis shows discernment in the '(Dis)Trust' environment, *X* axis shows discernment in the '(Dis)Like' environment. (**c**) Experiment 3 (N=391) showed the same results as Experiment 2. Data are plotted as box plots for each reaction, in which horizontal lines indicate median values, boxes indicate 25/75% interquartile range, and whiskers indicate 1.5 × interquartile range. Diamond shape indicates the mean discernment per reaction. Individuals' mean discernment data are shown separately as gray dots; symbols above each box plot indicate significance level compared to 0 using a t-test.***p < 0.001, **p < 0.01.

The online version of this article includes the following figure supplement(s) for figure 5:

**Figure supplement 1.** '(Dis)Trust' feedback improves discernment in sharing behavior (Experiments 5 and 6).

be disappointed with only a handful of 'hearts'. The user's goal is to maximize positive feedback per post. The same rationale as above holds for 'likes' and 'dislikes' except that those are less associated with veracity, thus impact discernment less.

The posts included in the experiment covered a range of topics including politics, science, health, environment, and society. As observed in *Figure 5b*, the effect of '(Dis)Trust' environment on discernment is observed regardless of content type.

Thus far, our results show that changing the incentive structure of social media platforms by coupling the number of 'carrots' and 'sticks' with information veracity could be a valuable tool to reduce the spread of misinformation. If feedback promotes discernment in sharing behavior, it is plausible that it may in turn improve belief accuracy. To test this, we asked participants at the end of the experiment to indicate how accurate they thought a post was on a scale from inaccurate (0) to accurate (100). Participants' error in estimating whether a post was true or false was calculated as follows: for false posts error was equal to the participants' accuracy rating and for true posts it was equal to 100 minus their rating. Participants' average error scores were entered into a between-subject ANOVA with type of feedback and valence of feedback, as well as political orientation and its interaction with feedback type. We observed an effect of type of feedback ($F(1,281) = 7.084$, p = 0.008, partial $\eta^2 = 0.025$), such that participants were more accurate (less errors) when they received '(Dis)Trust' feedback ($M = 47.24$, SE = 0.938) compared to '(Dis)Like' feedback ($M = 50.553$, SE = 0.851, $F(1,224) = 7.024$, p = 0.009, partial $\eta^2 = 0.03$). We further observed an effect of political orientation ($F(1,281) = 11.402$, p < 0.001, $\eta^2 = 0.039$), with Democrats ($M = 47.264$, SE = 0.773) being more accurate than Republicans ($M = 51.117$, SE = 0.802). No other effects were significant. All results hold when controlling for demographics, when not including political orientation in the analysis, and allowing for an interaction between type of feedback and valence (see *Supplementary file 8*). We replicated these results in Experiment 5. We again, observed an effect of feedback type ($F(1,258) = 4.179$, p = 0.042, partial $\eta^2 = 0.016$), such that participants were more accurate (less errors) when they received '(Dis)Trust' feedback ($M = 35.717$, SE = 0.65) compared to '(Dis)Like' feedback ($M = 37.63$, SE = 0.767; $F(1,212) = 3.955$, p = 0.048, partial $\eta^2 = 0.018$) and also more accurate than those who received no feedback (Baseline, $M = 39.73$, SE = 0.886; $F(1,162) = 11.759$, p < 0.001, partial $\eta^2 = 0.068$). No other effects were significant. These results hold when allowing for an interaction between type of feedback and valence (see *Supplementary file 9*).

## 'Trust' and 'distrust' incentives together improve discernment in sharing behavior (Experiment 3)

Given that Experiment 2 revealed that receiving 'trust' or 'distrust' feedback separately improves discernment, it is likely that the coupled presentation of both will jointly also improve discernment. To test this, we ran a third experiment with a new group of participants. The task was identical to Experiment 2 (see *Figure 4*), but this time we included three between-subject environments: a *Baseline* environment, in which participants received no feedback, a 'Trust & Distrust' environment, in which participants observed both the number of *trust* and the number of *distrust* feedback, and a 'Like & Dislike' environment, in which participants observed both the number of *like* and the number of *dislike* feedback.

Additionally, to ensure posts align equally with Democratic and Republican beliefs, in Experiment 3 we selected 40 posts (20 true, 20 false) in response to which Republicans and Democrats utilized the 'trust' button in a similar manner in Experiment 1 (see Materials and methods). Data of 18 participants were not analyzed according to pre-determined criteria (see Materials and methods for details). Analysis of Experiment 3 ($N = 391$, 194 Democrats, 197 Republican, $M_{age} = 35.304$, $SD_{age} \pm 11.089$; female = 196, male = 192, other = 3) was the same as in Experiment 2 except that there were three environments (Baseline, 'Like & Dislike', and 'Trust & Distrust') and no valence of feedback, because all environments either include both positive and negative feedbacks or no feedback.

Discernment was submitted to a between-subject ANOVA with *type of feedback* (Baseline/'Like & Dislike'/'Trust & Distrust'), political orientation and their interaction as factors. Again, we observed an effect of type of feedback ($F(1,385) = 11.009$, p < 0.001, partial $\eta^2 = 0.054$, *Figure 5c*), with participants in the 'Trust & Distrust' feedback group posting more true relative to false information ($M = 0.101$, SE = 0.015) than those in the 'Like & Dislike' group ($M = 0.042$, SE = 0.013; $F(1,261) = 8.478$, p = 0.00, partial $\eta^2 = 0.031$) or those who received no feedback at all ($M = 0.008$, SE = 0.014, $F(1,259)$

= 20.142, p < 0.001, partial $\eta^2$ = 0.0724). By contrast there was no difference between the latter two groups (F(1,250) = 2.981, p = 0.085, partial $\eta^2$ = 0.012). As observed in Experiment 2, Democrats (M = 0.073, SE = 0.011) were more discerning than Republicans (M = 0.031, SE = 0.012; F(1,385) = 6.409, p = 0.012, partial $\eta^2$ = 0.016). No other effects were significant.

Interestingly participants shared more frequently in the 'Trust & Distrust' environment compared to the other two environments (% of all trials: 'Trust & Distrust' = 36.2%, 'Like & Dislike' = 30.41%; Baseline = 25.853%; F(1,385) = 8.7692, p < 0.001, partial $\eta^2$ = 0.044). This illustrates that '(Dis)Trust' feedback improves discernment without reducing engagement. No other effects were significant.

All results hold when controlling for demographics, when not including political orientation in the analysis, and allowing for an interaction between type of reaction and valence (see *Supplementary files 10 and 11*). Results replicate in an independent sample (Experiment 6, see Materials and methods for details; and *Figure 5—figure supplement 1*).

At the end of Experiment 3, we again asked participants to indicate how accurate they thought a post was. Participants' average error scores were calculated as in Experiment 2 and entered into a between-subject ANOVA with type of feedback, political orientation and their interaction as factors. Democrats (M = 40.591; SE = 6.371) were more accurate than Republicans (M = 42.056; SE = 5.633; F(1,385) = 5.723, p = 0.017, partial $\eta^2$ = 0.015). No other effects were significant (for results controlling for demographics not including political orientation see *Supplementary file 12*). Note, that in the replication study (Experiment 6) we did observe an effect of type of feedback (F(1,147) = 4.596, p = 0.012, partial $\eta^2$ = 0.059), with '(Dis)Trust condition being most accurate. Thus, we see accuracy effects in three (Experiments 2, 5, and 6) out of our four experiments.

Taken together these findings suggest that changing the incentive structure of social media platforms, such that 'carrots' and 'sticks' are strongly associated with veracity promotes discernment in sharing behavior, thereby reducing the spread of misinformation.

## '(Dis)Trust' incentives improve discernment in sharing behavior by increasing the relative importance of evidence consistent with discerning behavior

Next, we set out to characterize the mechanism by which the new incentive structure increased discernment. Imagine you observe a post on social media, and you need to decide whether to share it – how do you make this decision? First, you examine the post. Second, you retrieve existing knowledge. For example, you may think about what you already know about the topic, what you heard others say, you may try to estimate how others will react to the post if you share it, and so on. This process is called 'evidence accumulation' – you gradually accumulate and integrate external evidence and internal evidence (memories, preferences, etc.) to decide. Some of the evidence you retrieve will push you toward a 'good' response that promotes veracity (i.e., posting a true post and skipping a false post) and some will push you toward a 'bad' response that obstructs veracity (i.e., posting a false post and skipping a true post). We can think of the evidence that pushes you toward a response that promotes veracity as 'signal'. Using computational modeling it is possible to estimate how much a participant is influenced ('pushed') by signal relative to noise, by calculating a parameter known as a 'drift rate' in a class of models known as drift-diffusion models (DDM). One possibility then is that in the '(Dis)Trust' environment evidence toward responses that promote veracity is given more weight than toward responses that obstruct veracity (i.e., the drift rate is larger), thus people make more discerning decisions.

Another, non-exclusive possibility, is that in the '(Dis)Trust' environment participants are more careful about their decisions. They require more evidence before making a decision. For example, they may spend more time deliberating the post. In DDM, this is estimated by calculating what is known as the distance between the decision thresholds (i.e., how much total distance do I need to be 'pushed' in one direction or the other to finally make a choice).

To test the above possible mechanisms, we modeled our data using the DDM (*Ratcliff, 1978*; *Ratcliff and McKoon, 2008*, see also *Lin et al., 2023*). We modeled participants' responses ('veracity-promoting' vs 'veracity-obstructing' choice) separately for each type of feedback ('(Dis)Trust', '(Dis)Like', Baseline) and each experiment (Experiments 2 and 3). The following parameters were included: (1) t(0) – the amount of non-decision time, capturing encoding and motor response time; (2) $\alpha$ – the distance between decision thresholds ('veracity-promoting' response vs 'veracity-obstructing'

**Table 1.** Group estimates for drift-diffusion model (DDM) in Experiment 2.

| Estimate | Baseline | '(Dis)Like' | '(Dis)Trust' |
|---|---|---|---|
| Distance between decision thresholds ($\alpha$) | 2.153<br>95% CI [2.09; 2.214] | 2.373<br>95% CI [2.281; 2.466] | 2.403<br>95% CI [2.280; 2.529] |
| Non-decision time ($t0$) | 7.025<br>95% CI [6.898; 7.154] | 6.936<br>95% CI [6.802; 7.071] | 6.681<br>95% CI [6.425; 6.94] |
| Starting point ($z$) | 0.497<br>95% CI [0.486; 0.508] | 0.491<br>95% CI [0.483; 0.50] | 0.48<br>95% CI [0.471; 0.49] |
| Drift rate ($v$) | 0.098<br>95% CI [0.039; 0.158] | 0.10<br>95% CI [0.056; 0.145] | 0.216<br>95% CI [0.17; 0.262] |

response); (3) $z$ – starting point of the accumulation process; and (4) $v$ – the drift rate (see Materials and methods for details; *Ratcliff, 1978*; *Ratcliff and McKoon, 2008*; *Voss et al., 2013*).

We next examined which of the parameters were different in the different environments (see *Tables 1 and 2*, and *Supplementary files 13 and 14* for highest density interval [HDI] comparisons). To that end, we calculated the difference in posterior distributions of each parameter for each pair of incentive environments ('(Dis)Trust' vs '(Dis)Like', '(Dis)Trust' vs Baseline, '(Dis)Like' vs Baseline) and report the 95% HDI of the difference. If the 95% HDI of the distribution does not overlap with zero, we infer a credible difference between the two incentive environments.

For both Experiment 2 (see *Figure 6a*) and Experiment 3 (see *Figure 6c*) we observed a meaningful difference in the drift rate. In particular, in the '(Dis)Trust' environments the drift rate was larger (Experiment 2: $v = 0.216$; Experiment 3: $v = 0.12$) than in the'(Dis)Like' environments (Experiment 2: $v = 0.01$; 95% HDI of difference [0.048; 0.183], Experiment 3: $v = 0.037$; 95% HDI of difference [0.032; 0.135]) or no feedback environment (Experiment 2: $v = 0.098$; 95% HDI of difference [0.041; 0.195]; Experiment 3: $v = 0.006$; 95% HDI of difference [0.061; 0.167]). The Baseline and '(Dis)Like' environments did not differ for drift rate (Experiment 2: 95% HDI of difference: [−0.075; 0.08]; Experiment 3: 95% HDI of difference [−0.016; 0.079]). This suggests that relative to the other environments, in the '(Dis) Trust' environments evidence consistent with a 'veracity-promoting' response is weighted more than 'evidence' consistent with a 'veracity-obstructing' response. We replicate these results in Experiments 5 and 6 (see *Supplementary files 15–18*).

While in Experiment 2 the decision threshold in the Baseline environment was lower than the other two environments, and non-decision time ($t0$) higher than in the '(Dis)Trust', these differences are not replicated in Experiment 3. More importantly, neither decision threshold nor non-decision time differed between the '(Dis)Trust' and '(Dis)Like' environments (see *Tables 1 and 2* and *Supplementary files 13 and 14* for HDI comparisons).

Model parameters could be successfully recovered with data simulated using group-level parameters from Experiments 2 and 3 separately (for details, see Materials and methods, see *Figure 6b, d*, *Supplementary files 19 and 20*). This was done by fitting the model to simulated data, in the same way as for the experimental data. We sampled 2000 times from the posteriors, discarding the first 500 as burn in. The same pattern of results was reproduced with the simulated data as with real participants' data (*Figure 7*). For each experiment, we ran two separate one-way ANOVAs to assess the effect of type of feedback on discernment: one for the real data and one for the simulated data. We

**Table 2.** Group estimates for drift-diffusion model (DDM) in Experiment 3.

| Estimate | Baseline | '(Dis)Like' | '(Dis)Trust' |
|---|---|---|---|
| Distance between decision thresholds ($\alpha$) | 2.238<br>95% CI [2.153; 2.328] | 2.207<br>95% CI [2.132; 2.286] | 2.209<br>95% CI [2.134; 2.286] |
| Non-decision time ($t0$) | 6.9<br>95% CI [6.762; 7.04] | 7.051<br>95% CI [6.918; 7.186] | 7.076<br>95% CI [6.944; 7.208] |
| Starting point ($z$) | 0.5<br>95% CI [0.49; 0.51] | 0.5<br>95% CI [0.49; 0.511] | 0.489<br>95% CI [0.476; 0.5] |
| Drift rate ($v$) | 0.006<br>95% CI [−0.027; 0.037] | 0.037<br>95% CI [0.002; 0.069] | 0.12<br>95% CI [0.086; 0.155] |

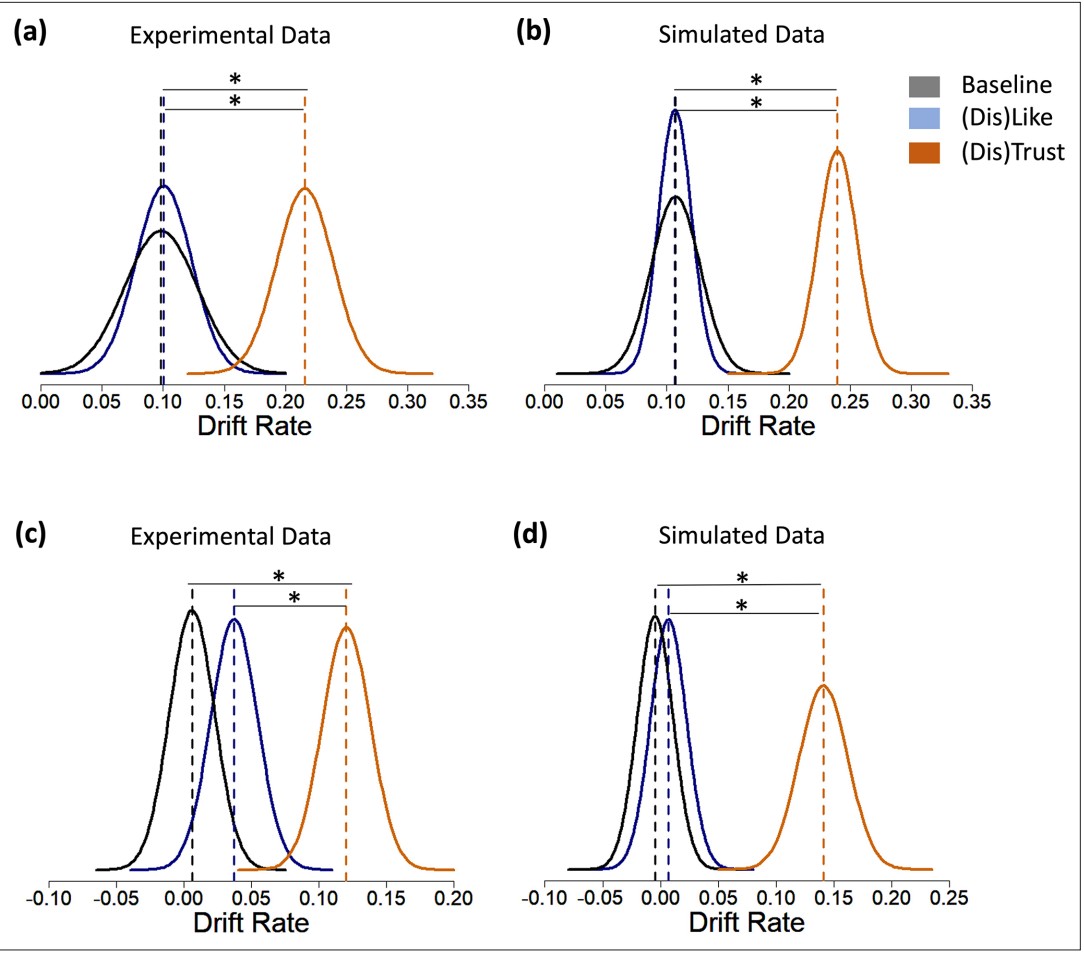

**Figure 6.** '(Dis)Trust' feedback increases the drift rate. Displayed are the posterior distributions of parameter estimates for the Baseline environment, the '(Dis)Like' environment and the '(Dis)Trust' environment. Dashed vertical lines indicate respective group means. In both (**a**) Experiment 2 (N=288) and (**c**) Experiment 3 (N=391) highest density interval (HDI) comparison revealed that participants had a larger drift rate in the '(Dis)Trust' environments than in the other environments. No credible difference was observed between the latter two environments. Recovered model parameter estimates reproduced experimental results for both (**b**) Experiment 2 and (**d**) Experiment 3. * indicates credible difference between environments.

remind the reader that we entered responses into our DDM as either 'veracity-promoting' (true post shared or false post skipped) or 'veracity-obstructing' (false post shared or true post skipped). Thus, discernment here is calculated as:

$$Discernment = Prop_{veracity-promoting\ responses} - Prop_{veracity-obstructing\ responses}$$

Which is equal to:

$$Discernment = Prop_{reposts\ true\ posts+skips\ false\ posts} - Prop_{reposts\ false\ posts+skips\ true\ posts}$$

As expected, we observed an effect of type of feedback in both the simulated (Experiment 2: $F(1,285) = 3.795$, p = 0.024, $\eta^2 = 0.026$; Experiment 3: $F(1,388) = 7.843$, p = 0.001, $\eta^2 = 0.039$, **Figure 7b, d**), and the experimental data (Experiment 2: $F(1,287) = 7.049$, p = 0.001, $\eta^2 = 0.047$; Experiment 3: $F(1,388) = 11.166$, p < 0.001, $\eta^2 = 0.054$). That is, discernment was higher in '(Dis)Trust' environments relative to '(Dis)Like' environments or no feedback environments (**Figure 7a, c**, see **Supplementary files 21 and 22** for pairwise comparisons and **Supplementary file 23** for correlations between real and recovered individual-level parameters).

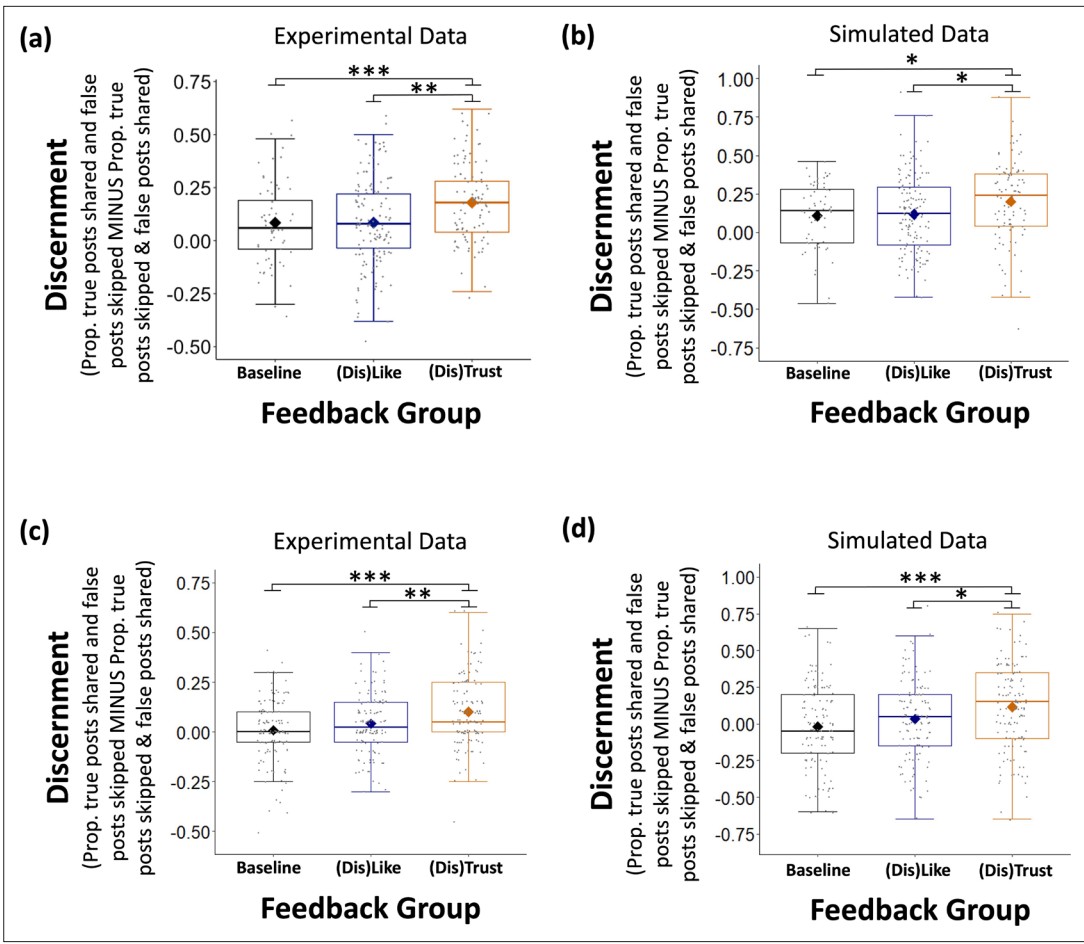

**Figure 7.** Simulated data reproduced experimental findings. One-way ANOVAs revealed that In both (**a**) Experiment 2 (N=288) and (**c**) Experiment 3 (N=391) participants who received '(Dis)Trust' feedback were more discerning than participants in the '(Dis)Like' and Baseline environments. Simulated data reproduced these findings (**b, d**). *Y* axis shows discernment, that is, proportion of true posts shared and false posts skipped minus the proportion of true posts skipped and false posts shared. *X* axis shows feedback group. Data are plotted as box plots for each reaction, in which horizontal lines indicate median values, boxes indicate 25/75% interquartile range and whiskers indicate 1.5 × interquartile range. Diamond shape indicates the mean discernment per reaction. Individuals' mean discernment data are shown separately as gray dots; symbols above each box plot indicate significance level compared to 0 using a t-test. ***p < 0.001, **p < 0.01, *p < 0.05.

## Discussion

Here, we created a novel incentive structure that significantly reduced the spread of misinformation and provide insights into the cognitive mechanisms that make it work. This structure can be adopted by social media platforms at no cost. The key was to offer reaction buttons (social 'carrots' and 'sticks') that participants were likely to use in a way that discerned between true and false information. Users who found themselves in such an environment, shared more true than false posts in order to receive more 'carrots' and less 'sticks'.

In particular, we offered 'trust' and 'distrust' reaction buttons, which in contrast to 'likes' and 'dislikes', are by definition associated with veracity. For example, a person may dislike a post about Joe Biden winning the election, however this does not necessarily mean that they think it is untrue. Indeed, in our study participants used 'distrust' and 'trust' reaction buttons in a more discerning manner than 'dislike' and 'like' reaction buttons. This created an environment in which the number of social rewards ('carrots') and punishments ('sticks') were strongly associated with the veracity of the information shared. Participants who were submitted to this new environment were more discerning in their sharing behavior compared to those in traditional environments who saw either no feedback

or 'dislike' and/or 'like' feedback. The result was a reduction in sharing of misinformation without a reduction in overall engagement. All the effects were replicated and effect sizes of misinformation reduction were large to medium.

Using computational modeling we were able to pin-point the changes to participants' decision-making process. In particular, drift-diffusion modeling revealed that participants in the new environment assigned more weight to evidence consistent with discerning than non-discerning behavior relative to traditional environments. In other words, the possibility of receiving rewards that are consistent with accuracy led to an increase in the weight participants assigned to accuracy-consistent evidence when making a decision. 'Evidence' likely includes external information that can influence the decision to share a post (such as the text and photo associated with the post) as well as internal information (e.g., retrieval of associated knowledge and memories).

Our results held when the potential feedback was only negative ('distrust'), only positive ('trust'), or both ('trust' and 'distrust'). While negative reaction buttons were used in a more discerning manner and more frequently than positive reaction buttons, we did not find evidence for a differential strength of positively or negatively valenced feedback on discernment of sharing behavior itself.

The findings also held across a wide range of different topics (e.g., politics, health, science, etc.) and a diverse sample of participants, suggesting that the intervention is not limited to a set group of topics or users, but instead relies more broadly on the underlying mechanism of associating veracity and social rewards. Indeed, we speculate that these findings would hold for different 'carrots' and 'sticks' (beyond 'trust' and 'distrust'), as long as people use these 'carrots' and 'sticks' to reward true information and punish false information. However, we speculate that the incentives were especially powerful due to being provided by fellow users and easily quantifiable (just as existing buttons including 'like' and 'heart'). This may contrast with incentives which are either provided by the platform itself and/or not clearly quantified such as verification marks (*Edgerly and Vraga, 2019*) or flagging false news (*Brashier et al., 2021*; *Chan et al., 2017*; *Grady et al., 2021*; *Lees et al., 2022*). Interestingly, a trust button has also been shown to increase sharing of private information (*Bălău and Utz, 2016*).

Finally, we observed that feedback not only promotes discernment in sharing behavior but may also increase the accuracy of beliefs. Though, while we see an increase in accuracy of beliefs in three of the four experiments, we did not observe this effect in Experiment 3. Thus, the new incentive structure reduces the spread of misinformation and may help in correcting false beliefs. It does so without drastically diverging from the existing incentive structure of social media networks by relying on user engagement. Thus, this intervention may be a powerful addition to existing intervention such as educating users on how to detect misinformation (e.g., *Lewandowsky and van der Linden, 2021*; *Maertens et al., 2021*; *Pilditch et al., 2022*; *Roozenbeek and van der Linden, 2019*; *Traberg et al., 2022*) or prompting users to think about accuracy before they engage in the platform (e.g., *Capraro and Celadin, 2022*; *Fazio, 2020*; *Pennycook and Rand, 2022*). Over time, these incentives may help users build better habits online (e.g., *Anderson and Wood, 2021*; *Ceylan et al., 2023*).

As real-world platforms are in the hands of private entities, studying changes to existing platforms requires testing simulated platforms. The advantage of this approach is the ability to carefully isolate the effects of different factors. However, 'real world' networks are more complex and involve additional features which may interact with the tested factors. Our hope is that the science described here will eventually impact how privately owned platforms are designed, which will reveal whether the basic mechanisms reported here hold in more complex scenarios.

This study lays the groundwork for integration of the new incentive structure into existing (and future) social media platforms to further test the external validity of the findings. Rather than removing existing forms of engagement, we suggest an addition that complements the existing system and could be adopted by social media platforms at no cost. The new structure could subsequentially help reduce violence, vaccine hesitancy and political polarization, without reducing user engagement.

## Materials and methods
### Experimental design
#### Power calculations
Sample sizes for all experiments were computed based on our pilot study (see Experiments 4–6). Power calculations were performed using g*Power (*Faul et al., 2009*) to achieve power of 0.8 (beta

= 0.2, alpha = 0.05; Experiment 1: partial $\eta^2$ = 0.51; Experiment 2: Cohen's $d$ = 0.33; Experiment 3: Cohen's $d$ = 0.327).

## Participants (Experiment 1)

One-hundred and eleven participants residing in the USA completed the task on *Prolific Academic*. Exclusion criteria were pre-established. Data of four participants who failed more than two memory checks were excluded from further analysis (see below). Thus, data of 107 participants were analyzed (52 Democrats, 54 Republican, 1 Other, $M_{age}$ = 40.579, $SD_{age}$ ± 14.512; female = 55, male = 52; Non-White = 20, White = 87). Participants received £7.50 per hour for their participation in addition to a memory test performance-related bonus. For all experiments presented in this article, ethical approval was provided by the Research Ethics Committee at University College London (#3990/003) and all participants gave informed consent. All experiments were performed in accordance with the principles expressed in the Declaration of Helsinki. All samples were politically balanced for Democrats and Republicans. All experiments were replicated using biological replicates (Experiments 4–6).

## Participants (Experiment 2)

Three-hundred and twenty participants completed the task on *Prolific Academic*. Data of four participants who failed more than two memory checks were excluded from further analysis (see below for details). Thus, data of 316 participants were analyzed (146 Democrats, 142 Republican, 28 Other, $M_{age}$ = 37.598, $SD_{age}$ ± 13.60; female = 157, male = 157, other = 2, Non-White = 77, White = 239). Participants received £7.50 per hour for the participation in addition to a memory test performance-related bonus.

## Participants (Experiment 3)

Four-hundred and nine participants completed the task on *Prolific Academic*. Data of three participants who failed more than two memory checks were excluded from further analysis (see Participants Experiment 1 for details). Further data of three participants who suspected that the feedback provided did not stem from real participants were excluded. Thus, data of four-hundred and three participants were analyzed (194 Democrats, 197 Republican, 12 Other, $M_{age}$ = 35.179, $SD_{age}$ ± 11.051; female = 204, male = 194, other = 4, Non-White = 85, White = 218). Participants received £7.50 per hour for their participation in addition to a memory test performance-related bonus.

## Participants (Experiment 4)

Fifty participants residing in the USA completed the task on *Prolific Academic* (25 Democrats, 8 Republican, 17 Other, $M_{age}$ = 33.16, $SD_{age}$ ±9.804; females = 24, male = 25, other = 1, Non-White = 15, White = 35). No participants failed the attention checks. Participants received £7.50 per hour for their participation in addition to a memory test performance-related bonus.

## Participants (Experiment 5)

Two-hundred and sixty-one participants completed the task on *Prolific Academic* (132 Democrats, 90 Republican, 39 Other, $M_{age}$ = 34.824, $SD_{age}$ ± 12.632; females = 122, males = 131, others = 8, Non-White = 84, White = 177). Participants received £7.50 per hour for their participation in addition to a memory test performance-related bonus.

## Participants (Experiment 6)

One-hundred and fifty participants completed the task on *Prolific Academic* (74 Democrats, 14 Republican, 62 Other, $M_{age}$ = 34.2, $SD_{age}$ ± 12.489; females = 70, males = 77, others = 3, Non-White = 39, White = 150). Participants received £7.50 per hour for their participation in addition to a memory test performance-related bonus.

## Task (Experiment 1)

Participants engaged in a simulated social media platform where they saw 100 news posts, each consisting of an image and a headline (see *Figure 2*, *Supplementary file 24* for stimuli and ratings). Half of the posts were true, and half were false. They covered a range of different topics including

COVID-19, environmental issues, politics, health, and society. They were all extracted from fact-checking website Politifact (https://www.politifact.com). For each post, participants had the option to either 'like', 'dislike', 'trust', or 'distrust' the post, or they could choose to 'skip' the post. They could press as many options as they wished (i.e., 'like' and 'distrust' for example). Participants were informed that if they chose to react to a post other users would be able to see their reactions. They were asked to treat the platform as they would any other social media network. The order in which reaction buttons appeared on screen was counterbalanced across participants. Participants also indicated their age, gender, ethnicity, and political orientation. The task was coded using the *Qualtrics* online platform (https://www.qualtrics.com).

## Memory/attention check

At the end of the experiment, participants were presented with five posts and had to indicate whether these were old or new. This is to ensure that participants were attentive during the experiment. Participants who failed more than two of the memory checks were excluded from the analysis.

## Task (Experiment 2)

In Experiment 2, participants engaged in a simulated social media platform where they saw the same 100 posts (50 true, 50 false) shown to participants in Experiment 1. Participants had to either 'repost' or 'skip' each post (see *Figure 4*). They were told that if they decided to repost, then the post would be shared to their feed, and they would observe other participants' reaction to it. We used a between-subject design with five environments. Depending on the environment participants were randomly assigned to, they could either see (1) how many people *disliked* the post, (2) how many people *liked* the post, (3) how many people *distrusted* the post, or (4) how many people *trusted* the post. We also included a *Baseline* environment, in which participants received no feedback. Due to logistic constraints, the feedback was not collected in real time but was instead taken from participants' reactions in Experiment 1. The participants, however, believed the reactions were provided in real time as indicated by a rotating cogwheel (1 s). If participants selected to skip, they would also observe a rotating cogwheel (1 s) and then a screen asking them to click continue. The average duration of the white screen ($M$ = 2.351 s; SE = 0.281) was not different from the average duration of feedback ($M$ = 2.625 s; SE = 0.245; $t$(233) = 0.853, p = 0.395, Cohen's $d$ = 0.056). Though the duration of trials in which participants chose to skip ($M$ = 9.046, SE = 0.38) was slightly shorter than those in which they chose to share ($M$ = 9.834, SE = 0.358; $t$(233) = 2.044, p = 0.042, Cohen's $d$ = 0.134). Thereafter, participants were presented with all the posts again and asked to indicate if they believed the post was accurate or inaccurate on a continuous scale from *0 = inaccurate* to *100 = accurate*. Finally, participants completed a short demographic questionnaire assessing age, gender, ethnicity, and political orientation. The task was self-paced. The task was coded using *JsPsych* and *Javascript*.

## Task (Experiment 3)

Experiment 3 (see *Figure 4*) was identical to the task used in Experiment 2 with three exceptions:

1. We selected 40 posts (20 true, 20 false), in which there was no significant difference in the way Republicans and Democrats reacted to them using the trust button during Experiment 1. This was done by entering participants' trust responses (0/1) into a vector for Democrats and Republicans for each post. We then performed Pearson chi-square tests for each of the 100 posts to identify whether Democrats and Republicans used the trust button differently. Posts where no significant difference was observed were included in Experiment 3.
2. Three environments were included: a *Baseline* environment, in which participants received no feedback, a 'Trust & Distrust' environment, in which participants received both *Trust* and *Distrust* feedback whenever they chose to share a post, and a 'Like & Dislike' environment, in which participants received *Like* and *Dislike* feedback whenever they chose to share a post.
3. At the end of the experiment, we asked participants: (1) 'What do you think the purpose of this experiment is?' (2) 'Did you, at any point throughout the experiment, think that the experimenter had deceived you in any way? If yes, please specify.'

## Task (Experiments 4–6)

The tasks and analysis in Experiments 4–6 were identical to those used in Experiments 1–3 except for the following differences:

1. In Experiment 4, a 'repost' button was included in addition to 'skip', '(Dis)Like', and '(Dis)Trust' options.
2. In Experiment 5, feedback symbols were colored – 'distrusts' and 'dislikes' in red and 'trusts' and 'likes' in green, instead of black and white.
3. Experiment 6 contained all 100 posts instead of a selection of 40 posts and did not contain final questions to assess whether participants believed the feedback was real.

## Statistical analysis

### Statistical analysis (Experiment 1)

We examined whether participants used the different reaction buttons to discern true from false information. For positive reactions (e.g., '*likes*' and '*trusts*') discernment is equal to the proportion of those reactions for true information minus false information, and vice versa for negative reactions ('*dislikes*' and '*distrusts*'). Proportions were calculated for each participant and then entered into a 2 (*type of reaction*: 'trust' and 'distrust'/'like' and 'dislike') by 2 (*valence*: positive, i.e., 'like', 'trust'/negative, i.e., 'dislike', 'distrust') within-subject ANOVA. Political orientation was also added as a between-subject factor (Republican/Democrat), allowing for an interaction of political orientation and type of reaction to assess whether participants with differing political beliefs used the reaction buttons in different ways. We performed one-sample *t*-tests to compare discernment (equal to the proportion of those reactions for true information minus false information, and vice versa for negative reactions) against zero to assess whether each reaction discerned between true and false information. To examine whether participants' frequency of use of each reaction option differed we again ran a within-subject ANOVA, but this time with percentage frequency of reaction option used as the dependent variable. We computed a Pearson's correlation across participants between frequency of skips and discernment.

One participant selected 'other' for political orientations. This participant was not included in the analysis because political orientation was included in analyses, and such small group sizes could heavily skew results. All statistical tests conducted in the present article are two sided. Analysis was conducted using IBM SPSS 27 and R Studio (Version 1.3.1056). All statistical tests conducted in the present article are two sided. All results of interest hold when controlling for demographics (age, gender, and ethnicity; see **Supplementary files 9–16**).

### Discernment analysis (Experiments 2 and 3)

Discernment is calculated for each participant by subtracting the proportion of sharing false information from the proportion of sharing true information. High discernment indicates greater sharing of true than false information. In Experiment 2, scores were submitted into an ANOVA with type of feedback ('(Dis)Trust' vs '(Dis)Like' vs Baseline), valence of feedback (positive, i.e., 'like', 'trust' vs negative, i.e., 'dislike', 'distrust'), political orientation and an interaction of political orientation and type of feedback. To assess whether frequency of posts shared differed we used the same ANOVA, this time with percentage of posts shared out of all trials as the dependent variable.

To test whether '(Dis)Trust' feedback improves belief accuracy, we transformed participants' belief ratings (which were given on a scale from post is accurate (100) or post is inaccurate (0)) to indicate error. If the post was false (inaccurate) error was equal to the rating itself, if the post was true (accurate) error was equal to 100 minus the rating. Participants' average error scores were then entered into a between-subject ANOVA with type of feedback (Baseline, '(Dis)Trust', '(Dis)Like'), valence of feedback, political orientation, and an interaction of political orientation and type of feedback.

Analysis of Experiment 3 followed that of Experiment 2 with the difference being that we had three type of feedback environments (Baseline, 'Like & Dislike', and 'Trust & Distrust') and of course no valence of feedback (as all environments were mixed valence or no valence). Data of participants who selected 'other' for political orientations (Experiment 2 = 28, Experiment 3 = 12) were not analyzed, because political orientation was included in the analyses variable, and small group sizes of 'other' could heavily skew results.

## Drift-diffusion modeling (Experiments 2 and 3)

To assess whether being exposed to an environment with '(Dis)Trust' feedback impacted the parameters of the evidence accumulation process in our data compared to Baseline and '(Dis)Like' feedback we analyzed our data using drift-diffusion modeling. To that end we ran three separate models – one for each type of feedback and included the following parameters: (1) $t(0)$, amount of non-accumulation/non-decision time; (2) $\alpha$, distance between decision thresholds; (3) $z$, starting point of the accumulation process; and (4) $v$, drift rate, is the rate of evidence accumulation.

We used the HDDM software toolbox (*Wiecki et al., 2013*) to estimate the parameters of our models. The HDDM package employs hierarchical Bayesian parameter estimation, using Markov chain Monte Carlo (MCMC) methods to sample the posterior probability density distributions for the estimated parameter values. We estimated both group- and individual-level parameters. Parameters for individual participants were assumed to be randomly drawn from a group-level distribution. Participants' parameters both contributed to and were constrained by the estimates of group-level parameters. In fitting the models, we used priors that assigned equal probability to all possible values of the parameters. Models were fit to log-transformed RTs. We sampled 20,000 times from the posteriors, discarding the first 5000 as burn in and thinning set at 5. MCMCs are guaranteed to reliably approximate the target posterior density as the number of samples approaches infinity. To test whether the MCMC converged within the allotted time, we used Gelman–Rubin statistic (*Gelman and Rubin, 1997*) on five chains of our sampling procedure. The Gelman–Rubin diagnostic evaluates MCMC convergence by analyzing the difference between multiple Markov chains. The convergence is assessed by comparing the estimated between- and within-chain variances for each model parameter. In each case, the Gelman–Rubin statistic was close to one (<1.1), suggesting that MCMC were able to converge.

We then compared parameter estimates using 95% HDI. Specifically, for each comparison ('(Dis) Trust' vs '(Dis)Like', '(Dis)Trust' vs Baseline, '(Dis)Like' vs Baseline) we calculated the difference in the posterior distributions and reported the 95% HDI of the difference. If this HDI did not include zero, we consider there to be a meaningful difference between the two feedback types compared. To validate the winning model, we used each group's parameters obtained from participants' data to simulate log-transformed response times and responses separately for each feedback type. We used the exact number of subjects and number of trials as in the experiments. Simulated data were then used to (1) perform model recovery analysis and (2) to compare the pattern of participants' response to the pattern of simulated responses, separately for each group. We sampled 2000 times from the posteriors, discarding the first 500 as burn in. Simulation and model recovery analysis were performed using the HDDM software toolbox (*Wiecki et al., 2013*). One-way ANOVAs were computed to examine if simulated data reproduced the behavioral pattern from experimental data. To that end, discernment was entered into a one-way ANOVA with type of feedback as the independent variable for Experiments 2 and 3 separately. Note, that as we did not enter veracity of the post into our DDM and instead entered responses as either 'veracity-promoting' (true post shared or false post skipped) or 'veracity-obstructing' (false post shared or true post skipped). Thus, discernment was calculated as the proportion of true posts shared and false posts skipped minus the proportion of true posts skipped and false posts shared.

## Analysis (Experiments 4–6)

The analyses were identical to those in Experiment 1–3 except that the samples were not politically balanced (see Participants Experiments 4–6), as such analysis did not take into account political orientation.

# Acknowledgements

We thank Valentina Vellani, Bastien Blain, Irene Cogliati Dezza, Moshe Glickman, Sarah Zheng, India Pinhorn, Christopher Kelly, and Hadeel Haj-Ali for comments on previous versions of the manuscript. TS is funded by a Wellcome Trust Senior Research Fellowship 214268/Z/18/Z.

# Additional information

## Funding

| Funder | Grant reference number | Author |
|---|---|---|
| Wellcome Trust | 214268/Z/18/Z | Tali Sharot |

The funders had no role in study design, data collection and interpretation, or the decision to submit the work for publication. For the purpose of Open Access, the authors have applied a CC BY public copyright license to any Author Accepted Manuscript version arising from this submission.

## Author contributions

Laura K Globig, Conceptualization, Formal analysis, Investigation, Visualization, Methodology, Writing – original draft, Project administration, Writing – review and editing; Nora Holtz, Conceptualization, Project administration; Tali Sharot, Conceptualization, Supervision, Funding acquisition, Writing – original draft, Writing – review and editing, Methodology

## Author ORCIDs

Laura K Globig (ID) http://orcid.org/0000-0002-0612-0594
Tali Sharot (ID) http://orcid.org/0000-0002-8384-6292

## Ethics

For all experiments presented in this article, ethical approval was provided by the Research Ethics Committee at University College London and all participants gave informed consent (#3990/003).

## Decision letter and Author response

Decision letter https://doi.org/10.7554/eLife.85767.sa1
Author response https://doi.org/10.7554/eLife.85767.sa2

---

# Additional files

## Supplementary files

- Supplementary file 1. Discernment of reactions (Experiment 1).
- Supplementary file 2. % Reactions out of all posts (Experiment 1).
- Supplementary file 3. Discernment of reactions (Experiment 4, including type x valence of reaction interaction).
- Supplementary file 4. % true and false posts shared out of all true or false posts in that feedback condition (Experiment 2).
- Supplementary file 5. Discernment of sharing behavior (Experiment 2).
- Supplementary file 6. % posts shared out of all posts (Experiment 2).
- Supplementary file 7. Discernment of sharing behavior (Experiment 5).
- Supplementary file 8. Belief Accuracy (Experiment 2).
- Supplementary file 9. Belief Accuracy (Experiment 5).
- Supplementary file 10. Discernment of sharing behavior (Experiment 3).
- Supplementary file 11. % posts shared out of all posts (Experiment 3).
- Supplementary file 12. Belief Accuracy (Experiment 3).
- Supplementary file 13. Mean difference in posterior distributions and 95% HDI Comparison (Experiment 2).
- Supplementary file 14. Mean difference in posterior distributions and 95% HDI Comparison (Experiment 3).
- Supplementary file 15. Group estimates for DDM (Experiment 5).
- Supplementary file 16. Mean difference in posterior distributions and 95% HDI Comparison (Experiment 5).
- Supplementary file 17. Group estimates for DDM (Experiment 6).

• Supplementary file 18. Mean difference in posterior distributions and 95% HDI Comparison (Experiment 6).

• Supplementary file 19. Recovered Group estimates for DDM based on simulated data (Experiment 2).

• Supplementary file 20. Recovered Group estimates for DDM based on simulated data (Experiment 3).

• Supplementary file 21. Pairwise Comparisons for Discernment (Experiment 2).

• Supplementary file 22. Pairwise Comparisons for Discernment (Experiment 3).

• Supplementary file 23. Correlations between participants' real and recovered DDM estimates (Experiment 2 and Experiment 3).

• Supplementary file 24. Individual Ratings per Stimulus (Experiment 1).

• MDAR checklist

## Data availability

Code and anonymized data are available at: https://github.com/affective-brain-lab/Changing-the-Incentive-Structure-of-Social-Media-Platforms (copy archived at *Globig, 2023*).

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
