## [Editor Report]

This important paper outlines a novel method for reducing the spread of misinformation on social media platforms. A compelling series of experiments and replications support the main claims, which could have significant real-world societal impact.

---

## [Decision Letter]

**Decision letter after peer review:**

Thank you for submitting your article "Changing the Incentive Structure of Social Media Platforms to Halt the Spread of Misinformation" for consideration by *eLife* – we enjoyed reading it and think it could have significant real-world impact.

Your article has been reviewed by 3 peer reviewers, including Claire M Gillan as the Reviewing Editor and Reviewer #1, and the evaluation has been overseen by Michael Frank as the Senior Editor. The reviewers have discussed their reviews with one another, and the Reviewing Editor has drafted this to help you prepare a revised submission.

Essential revisions:

1) Clarity and consistency of analysis. While all the reviewers applauded the replications and comprehensiveness of the study, it was felt that some important analyses were omitted. Of note, the valence by response type interaction was not carried out, though there were valence effects. There are also queries about the use of political orientation in these main analyses given it is not the focus of the paper.

2) More detail of the design of the study, including the instructions provided to participants and reporting of the ratings of trust/distrust, like/dislike from study 1 (which are used for study 2). Means are reported, but distributions of ratings for each post would be helpful to understand the feedback received in study 2.

3) The DDM is not essential to the main results, but there were questions about the omission of the starting bias and in general the methods could have been better fleshed out.

4) Although some reviewers felt the question was more applied, it was suggested there be more emphasis on theoretical motivations for the study. So you might consider striking a balance here.

*Reviewer #1:*

This is a very comprehensive and compelling piece of research that investigates a novel method for reducing the spread of misinformation online. The authors introduce trust/distrust options into a (analogue) social media environment and illustrate that it improves discernment (sensitivity of users to the factual basis of content they share). Pairing cross-sectional, correlational designs with causal manipulation and replication across multiple independent studies, this is a tour de force. There is immediate real-world impact from the work; the data present a compelling case that misinformation would be spread less often if social media environments included peer-assessments of the trustworthiness of content. The authors additionally showed that trust/distrust options actually increase engagement with social media over the existing like/dislike buttons, which creates an interesting additional incentive for private companies in this space to implement this. Finally, they showed that these social incentives around trust actually increase belief accuracy in users.

The authors compliment these insights with mechanistic explanations by modelling the 'sharing of posts' data using a drift diffusion model and comparing parameters across environments where trust/distrust vs like/dislike were differentially available. They find that across two studies drift rates are larger in the trust/distrust conditions, suggesting users accumulate more information before making a decision.

The authors note that the complexities of real world social media environments may work differently to the tightly controlled laboratory analogues described here and this would be an important next step. Other papers have investigated methods to promote the sharing of true information online, but they are typically active training conditions that would be difficult to roll out and achieve engagement. In contrast, the approach here is naturalistic and gamified in the same sense like/dislikes are on these platforms and demonstrably increases engagement.

I really have no major suggestions at all. The paper is very clear and extremely comprehensive. It builds nicely on this recent PNAS paper (https://www.pnas.org/doi/epdf/10.1073/pnas.2216614120), though no need to cite it necessarily, it highlights the need for methods that can promote discernment via their reward systems – the present paper appears to do just that.

*Reviewer #2:*

This study presents important data that changing the incentive structure in social platform can decrease the spread of misinformation. If implemented in the real world, these findings may help solve an important problem in our society. The evidence supporting the conclusions is solid but more clarity in the way the analyses were conducted would strengthen the study.

One strength of the study is that it tests a clear hypothesis, namely that changing the incentive structure in social media platforms (so that rewards and punishment that users receive reflect the veracity of the information they share) will decrease the spread of misinformation. The authors suggest changing the current like/dislike feedback by trust/distrust feedback, the latter implying a judgement about veracity of the statement.

The study involves 3 different experiments. One strength of the study is that the main fundings are replicated in 3 independent experiments. In the first experiment, the authors demonstrate that participants discern true from false information when deciding to trust or distrust a new post to a higher degree than when deciding to like or dislike a post. This claim seems to be supported by the data. In the second and the third studies, the authors show that participants are more likely to share true information when their decisions to share a post result in trust or distrust feedback in comparison to like or dislike feedback or no feedback at all. Moreover, seeing how other participants trust/distrust the posts increase the accuracy of the beliefs that participants hold at the end of the experiment. These claims also seem to be supported by the data.

These results are clearly relevant and provide a clear and easy to implement solution to a very hot problem in our contemporary society, namely spread of misinformation via social media that contributes to polarization and consolidation of conspiracy theories. In this sense, these results might be very valuable. However, I missed information about what participants were told about the experiment and what were they instructed to do. This information has implications to understand the possible generalization of the results outside of the experimental setting. For example, the authors show that participants engage more with the trust/distrust than with the like/dislike buttons. It is not clear how this result may be influenced by the fact that participants are doing an experiment about trustworthiness of social media posts. The authors rightly acknowledge that their simulated social network is a simplification of a more complex real system. Hence, the importance of knowing every detail of the experimental paradigm and instructions given to the participants.

The authors used DDM to understand the underlying cognitive mechanism of the observed effects. The standard DMM as implemented in this study assumes that the decision-making process is at equilibrium and does not account for changes due to learning during the experiment. This is a drawback considering that the main effect of changing the incentive structure is expected to be achieved through a learning process. Nevertheless, the DDM could still disentangle between alternative cognitive/computational explanations of the differences between the experimental conditions. But the current implementation is difficult to fully interpret. It is unclear why the starting point is fixed and not a free parameter. Theoretically, it is equally likely that the experimental manipulation induces a bias towards discernment in repost behavior. And if there is a good reason to fix the starting point, it is unclear how it was chosen and how this decision impacts the rest of the DMM results.

Although strongly mitigated by the fact that the main results are replicated, one possible drawback of the study is that it was not preregistered. The analysis pipeline involves a few researchers' degrees of freedom. I list 3 examples:

1) It is unclear why the ANOVA in experiment 1 did not include an interaction between type of response and valence. The data seems to show a pattern towards this direction whereby the main behavioral effect is driven by a difference between dislike and distrust whereas the difference between like and trust seems much smaller or non-significant. The omission of the interaction might have affected the reported results.

2) I am not sure that the use of political orientation in the analysis is fully adequate. The authors report an interaction with the main effect in experiment 1 driven by stronger results among democrats when compared to republicans. However, in experiment 2, the interaction is only marginally significant. Despite this, the authors still explore it further to show a difference in discernment between democrats and republicans. In experiment 3, the authors only used a subset of posts that were equally trusted by democrats and republicans. Importantly, the data shows that effect of trust/distrust on repost discernment is unaffected, which strengthens the main results without doubt. However, this rises the possibility that the interaction in experiment 1 is driven by the selection of material. It is also unclear to what extend the results in experiment 1 and 2 are dependent on the inclusion of political orientation in the models. Also, the inclusion of this factors justifies the exclusion of a significant number of participants in experiment 2 and 3.

3) In experiment 2 and 3, the main results are analyzed using a GLM approach. It is unclear why an ANOVA is not used here and how the main regressor was built. The latter may have implications for interpretation of the results.

Try to increase clarity in some points of the text. The use of the term discernment in the abstract is not clear as the term is only defined in the text. State clearly in the Results section that experiment 2 and 3 are between subject (now it is only stated in the figure) along with the number of participants per group, including the number of excluded participants.

Include the interaction between type of response and valence in study 1. Report the results with and without political orientation to show that the results are independent of the inclusion of this factor.

I would suggest that the DDM analysis either includes all model parameters as free or that the authors try to implement different variations (with different free parameters) and perform model comparison to test which is the most parsimonious fit to the data. In the recoverability analysis, I would report the correlation between the simulated and recovered parameters. And if different models are tested, I would include a confusion matrix of the recoverability of the models.

When presenting the DDM results, I would avoid the systematic use of "good" and "bad" labels for the responses. It is fine to use this metaphor in one instance, but it sounds like an oversimplification when it is repeated many times.

In the first paragraph of the discussion, the authors state "Users who found themselves in such an environment began sharing more true than false posts in order to receive carrots and avoid sticks." I would reformulate this statement not to imply learning. Although learning is implicit by the fact that the effects are a consequence of changing the incentive structure, the behavioral data analysis is aggregated across the whole experiment.

Figure 7 is missing

*Reviewer #3:*

This paper suggests that changing the incentive structure on social platforms may influence people's discernment in their sharing decisions. Particularly, in experiment 1, the authors suggest that participants' use of trust-distrust buttons were more discerning than their use of like-dislike button. In experiments 2 and 3, the authors suggest that operating in a trust-distrust environment makes people's sharing more discerning than operating in a like-dislike or no-feedback environments. They also test a drift rate and claim that people place more weight to accuracy when provided with the distrust-trust feedback.

Strengths:

– The paper examines a novel mechanism to prevent misinformation spread on social platforms.

– The authors' idea of offering buttons that are better linked to headline veracity is a practical solution that could be easily implemented.

Weaknesses:

– The conceptualization could be better articulated with wider coverage of the misinformation literature.

– There is a disconnect between the theory and the empirical studies.

– Fake and true headlines may be different from one another on other factors (e.g., emotions) that affect people's trust judgments.

– The design of experiments 2 and 3 fails to accurately connect trust judgments to the veracity of the headlines.

Major Comments

Conceptualization:

Conceptually, authors suggest that adding buttons should make people's evaluations of headlines (in experiment 1) and the presence of much wider variety of feedback will make their sharing decisions better (in experiments 2 and 3).

1. The reader would like to see better build-up of the theoretical framework. Some questions are:

a. Whereas social media sharing (especially sharing 40-100 headlines) seem to be repetitive and can lead participants to develop habits (Anderson and Wood 2021; Ceylan et al., 2023). Do habits play a role? With the change in incentive structure, do participants develop different sharing habits?

b. Would we expect differential evaluation and sharing as a function of valence of the feedback button? The authors claim in many places that "… humans naturally seek social rewards and are motivated to avoid social punishments (Behrens, Hunt, Woolrich, and Rushworth, 2008; Bhanji and Delgado, 2014)." Do they test and find evidence for this? (More discussion and suggestions are below).

2. Figure 1b and 1c are rather confusing than clarifying. If I understand correctly, in experiment 2, participants receive one type of feedback that is disconnected from the actual veracity of the information. If people receive just distrust feedback for all the headlines (true and false), how is this in line with the idea that authors align carrots with truth and sticks with falsity?

Empirics:

My overall feedback is that authors should better connect their conceptualization to their empirics.

3. Why is the participant the evaluator of the news headline in experiment 1 and then the sharer of the headline in experiments 2 and 3?

4. The reader wonders where the action comes from: is it from reduction in rejecting false information or increasing in endorsing true information? In the model authors shared in the Supplemental Appendix Table 2, can the authors run a mixed effect model on the % of headlines engaged (not skipped) as a function of type of reaction and valence of reaction? This will help authors test their theory more robustly.

a. The same for Supplemental Appendix Tables 5 and 7. Authors can test their theory by including an interaction term for type of reaction and valence of that reaction.

5. In experiment 1, how did "skip" reaction impact people's discernment? Were those who skipped more discerning than those who used like-dislike buttons?

6. In experiments 2 and 3, the authors provide participants with 100 headlines. Participants make a skip or repost decision. Then, participants receive one of the types of feedback depending on their condition. One problem with this design and what authors suggest in Figure 1 is that participants receive one type of feedback in their respective condition. For instance, in the distrust condition, participants receive distrust feedback no matter what the veracity of the headline is. This is problematic because in this model, people do not have an opportunity to learn and connect veracity with trust feedback. I suggest authors could collect new data by cleaning up this mapping. The reader wonders how the results would change if this mapping actually happens and participants receive trust feedback when they share accurate information and distrust feedback when they share false information.

a. This may explain why the authors do not find a differential effect between trust and distrust conditions. Trust feedback that is disconnected from the actual veracity of the headline would not enhance people's discernment differentially when this feedback is negative or positive. It seems like participants were just operating in an environment knowing that the feedback will be about trust but no matter what they share the valence of the trust would not change.

7. Further, in the current execution, participants learn about the reward and use this learning in their sharing decision all at one stage. However, the execution of the study can be cleaner if the authors separate the learning stage from the test stage (see Study 4 in Ceylan et al., 2023).

8. One concern is that the authors stripped the headlines from any factor that could contribute people's truth and trust judgments. For instance, source credibility or consensus are factors that aid people's judgments on social platforms such as Facebook and Twitter. One may wonder how people make their truth judgments in the absence of these cues. Can people discern true information from false information? In experiment 2, authors measure the discernment, but I did not see the means between true and false headlines. Also, do people's trust and like responses correlate with the actual veracity of the information?

9. Trust judgments also bear on other factors. One such factor is emotions (Dunn and Schweitzer, 2005). The reader wonders what the valence of the emotions is between the true and false headlines. A pretest that shows differences in emotions between false and true headlines would be helpful to the reader.

10. I am not very familiar with the computational analysis calculating drift rate. For readers who will not be very familiar with this method will need more detail about the analysis. Also, it seems like the results from experiments 2 and 3 do not show similar trends. Specifically, in experiment 2, the distance between the thresholds is the highest in the trust-distrust conditions and the non-decision time is the lowest. In experiment 3, there is no difference in distance between the thresholds across all three conditions. This time, the non-decision time was the highest in the trust-distrust conditions. How do the authors reconcile these differences?

References

Anderson, I. A., and Wood, W. (2021). Habits and the electronic herd: The psychology behind social media's successes and failures. Consumer Psychology Review, 4(1), 83-99.

Ceylan, G., Anderson, I. A., and Wood, W. (2023). Sharing of misinformation is habitual, not just lazy or biased. Proceedings of the National Academy of Sciences, 120(4), e2216614120.

Dunn, J. R., and Schweitzer, M. E. (2005). Feeling and believing: the influence of emotion on trust. Journal of personality and social psychology, 88(5), 736.

---

## [Author Response]

Essential revisions:1) Clarity and consistency of analysis. While all the reviewers applauded the replications and comprehensiveness of the study, it was felt that some important analyses were omitted. Of note, the valence by response type interaction was not carried out, though there were valence effects. There are also queries about the use of political orientation in these main analyses given it is not the focus of the paper.

We thank the reviewers for this positive evaluation of our study. We have repeated all the main analyses including valence and type of reaction/feedback interactions, as well as not including political orientation as a factor. The results show once again that discernment is greater in the ‘(Dis)Trust’ condition than the other conditions (Supplementary files 1-12).

2) More detail of the design of the study, including the instructions provided to participants and reporting of the ratings of trust/distrust, like/dislike from study 1 (which are used for study 2). Means are reported, but distributions of ratings for each post would be helpful to understand the feedback received in study 2.

We now provide more details pertaining to the design of the study. In particular, we now clarify the between-subject design nature of Experiment 2 and 3 (pg.13, 21, 35-36) and provide the full instructions for all studies (Figure 2 – —figure supplement 1, Figure 4 – —figure supplement 1 and 2). We also report the exact ratings for each post in Experiment 1 (Supplementary file 24).

3) The DDM is not essential to the main results, but there were questions about the omission of the starting bias and in general the methods could have been better fleshed out.

We now follow the reviewer’s recommendation and include the starting point as a free parameter (see pg. 23-28, Figure 6 and 7, Tables 1-2). The findings remain the same. In particular we find that the drift rate in the ‘(Dis)Trust’ environments (Experiment 2: v=0.216; Experiment 3: v=0.12) was meaningfully higher than the drift rate in both the ‘(Dis)Like’ (Experiment 2: v=0.01; 95% HDI of difference [0.048; 0.183], Experiment 3: 0.034; 95% HDI of difference [0.033;0.139]) and Baseline environments (Experiment 2: v=0.098; 95% HDI of difference [0.041; 0.195]; Experiment 3: v=0.007; 95% HDI of difference [0.062; 0.165]).

4) Although some reviewers felt the question was more applied, it was suggested there be more emphasis on theoretical motivations for the study. So you might consider striking a balance here.

Thank you for the suggestion, we have now added more details regarding the theory and rationale on pg. 4-7, 16-17, 30-31.

Reviewer #1:This is a very comprehensive and compelling piece of research that investigates a novel method for reducing the spread of misinformation online. The authors introduce trust/distrust options into a (analogue) social media environment and illustrate that it improves discernment (sensitivity of users to the factual basis of content they share). Pairing cross-sectional, correlational designs with causal manipulation and replication across multiple independent studies, this is a tour de force. There is immediate real-world impact from the work; the data present a compelling case that misinformation would be spread less often if social media environments included peer-assessments of the trustworthiness of content. The authors additionally showed that trust/distrust options actually increase engagement with social media over the existing like/dislike buttons, which creates an interesting additional incentive for private companies in this space to implement this. Finally, they showed that these social incentives around trust actually increase belief accuracy in users.The authors compliment these insights with mechanistic explanations by modelling the 'sharing of posts' data using a drift diffusion model and comparing parameters across environments where trust/distrust vs like/dislike were differentially available. They find that across two studies drift rates are larger in the trust/distrust conditions, suggesting users accumulate more information before making a decision.The authors note that the complexities of real world social media environments may work differently to the tightly controlled laboratory analogues described here and this would be an important next step. Other papers have investigated methods to promote the sharing of true information online, but they are typically active training conditions that would be difficult to roll out and achieve engagement. In contrast, the approach here is naturalistic and gamified in the same sense like/dislikes are on these platforms and demonstrably increases engagement.

We thank the reviewer for this positive evaluation.

I really have no major suggestions at all. The paper is very clear and extremely comprehensive. It builds nicely on this recent PNAS paper (https://www.pnas.org/doi/epdf/10.1073/pnas.2216614120), though no need to cite it necessarily, it highlights the need for methods that can promote discernment via their reward systems – the present paper appears to do just that.

We thank the reviewer for this positive evaluation.

Reviewer #2:This study presents important data that changing the incentive structure in social platform can decrease the spread of misinformation. If implemented in the real world, these findings may help solve an important problem in our society. The evidence supporting the conclusions is solid but more clarity in the way the analyses were conducted would strengthen the study.One strength of the study is that it tests a clear hypothesis, namely that changing the incentive structure in social media platforms (so that rewards and punishment that users receive reflect the veracity of the information they share) will decrease the spread of misinformation. The authors suggest changing the current like/dislike feedback by trust/distrust feedback, the latter implying a judgement about veracity of the statement.The study involves 3 different experiments. One strength of the study is that the main fundings are replicated in 3 independent experiments. In the first experiment, the authors demonstrate that participants discern true from false information when deciding to trust or distrust a new post to a higher degree than when deciding to like or dislike a post. This claim seems to be supported by the data. In the second and the third studies, the authors show that participants are more likely to share true information when their decisions to share a post result in trust or distrust feedback in comparison to like or dislike feedback or no feedback at all. Moreover, seeing how other participants trust/distrust the posts increase the accuracy of the beliefs that participants hold at the end of the experiment. These claims also seem to be supported by the data.These results are clearly relevant and provide a clear and easy to implement solution to a very hot problem in our contemporary society, namely spread of misinformation via social media that contributes to polarization and consolidation of conspiracy theories. In this sense, these results might be very valuable.

We thank the reviewer for this summary of the findings and for their positive assessment.

However, I missed information about what participants were told about the experiment and what were they instructed to do. This information has implications to understand the possible generalization of the results outside of the experimental setting. For example, the authors show that participants engage more with the trust/distrust than with the like/dislike buttons. It is not clear how this result may be influenced by the fact that participants are doing an experiment about trustworthiness of social media posts. The authors rightly acknowledge that their simulated social network is a simplification of a more complex real system. Hence, the importance of knowing every detail of the experimental paradigm and instructions given to the participants.

We thank the reviewer for prompting us to provide the exact instructions. These are now available in full in Figure 2 – —figure supplement 1, Figure 4 – —figure supplement 1 and 2.

The authors used DDM to understand the underlying cognitive mechanism of the observed effects. The standard DMM as implemented in this study assumes that the decision-making process is at equilibrium and does not account for changes due to learning during the experiment. This is a drawback considering that the main effect of changing the incentive structure is expected to be achieved through a learning process. Nevertheless, the DDM could still disentangle between alternative cognitive/computational explanations of the differences between the experimental conditions. But the current implementation is difficult to fully interpret. It is unclear why the starting point is fixed and not a free parameter. Theoretically, it is equally likely that the experimental manipulation induces a bias towards discernment in repost behavior. And if there is a good reason to fix the starting point, it is unclear how it was chosen and how this decision impacts the rest of the DMM results.

Following the reviewer’s recommendation, we now include the starting point as a free parameter. Again, we find that the drift rate in the ‘(Dis)Trust’ environments (Experiment 2: v=0.216; Experiment 3: v=0.12) was meaningfully higher than the drift rate in both the ‘(Dis)Like’ (Experiment 2: v=0.01; 95% HDI of difference [0.048; 0.183], Experiment 3: 0.037; 95% HDI of difference [0.032;0.135]) and Baseline environments (Experiment 2: v=0.098; 95% HDI of difference [0.041; 0.195]; Experiment 3: v=0.006; 95% HDI of difference [0.061; 0.167]). We do not find a difference in starting point bias across conditions in Experiment 2 and 3 (see pg. 23-28, Figure 6 and 7, Tables 1-2), though in the replications studies, the baseline starting point is greater than in the ‘(Dis)Trust’ and/or ‘(Dis)Like’ conditions (see Supplementary files 15-18).

Although strongly mitigated by the fact that the main results are replicated, one possible drawback of the study is that it was not preregistered. The analysis pipeline involves a few researchers' degrees of freedom. I list 3 examples:1) It is unclear why the ANOVA in experiment 1 did not include an interaction between type of response and valence. The data seems to show a pattern towards this direction whereby the main behavioral effect is driven by a difference between dislike and distrust whereas the difference between like and trust seems much smaller or non-significant. The omission of the interaction might have affected the reported results.

Following the reviewer’s suggestion, we have now added the valence x type of reaction interaction to the analysis of Experiment 1. Once again, we find a main effect of type of reaction (F(1,106)=80.936, p<0.001, partial η2=0.43) and valence (F(1,106)=18.26, p<0.001, partial η2=0.15). There was also an interaction of valence x type of reaction (F(1,106)=51.489, p<0.001, partial η2=0.33), which was characterized by the ‘distrust’ button (M=0.157, SE=0.008) being used in a more discerning manner than the ‘trust’ button (M=0.099, SE=0.008; t(106)=9.338, p<0.001, Cohen’s d=0.903), while the ‘like’ button (M=0.06, SE=0.008) was used in a more discerning manner than the ‘dislike’ button (M=0.034, SE=0.008; t(106)=3.474, p<0.001, Cohen’s d=0.336). These results are reported in Supplementary file 1.

2) I am not sure that the use of political orientation in the analysis is fully adequate. The authors report an interaction with the main effect in experiment 1 driven by stronger results among democrats when compared to republicans. However, in experiment 2, the interaction is only marginally significant. Despite this, the authors still explore it further to show a difference in discernment between democrats and republicans. In experiment 3, the authors only used a subset of posts that were equally trusted by democrats and republicans. Importantly, the data shows that effect of trust/distrust on repost discernment is unaffected, which strengthens the main results without doubt. However, this rises the possibility that the interaction in experiment 1 is driven by the selection of material. It is also unclear to what extend the results in experiment 1 and 2 are dependent on the inclusion of political orientation in the models. Also, the inclusion of this factors justifies the exclusion of a significant number of participants in experiment 2 and 3.

Following the reviewer’s comment, we repeated all our analyses without including political orientation as a factor and thus not excluding participants who were ‘independent’. Again, we find in all studies that discernment is greater in the ‘(Dis)Trust’ conditions than the other conditions (Supplementary files 1-12).

We now also clarify our motivation for including political orientation in the main text. In particular, we included political orientation in our analysis because previous studies suggest a political asymmetry in both the sharing of misinformation online (e.g., Grinberg et al., 2019; Guess et al., 2019) as well as in the subsequent efficacy of interventions aimed at reducing misinformation (e.g., Roozenbeek et al., 2020; Pennycook et al., 2021). For an intervention to be effective it is helpful that it would work across both sides of the political spectrum. As such it is common practice to include political orientation in analyses assessing the efficacy of novel interventions (e.g., Roozenbeek et al., 2022, Pennycook and Rand, 2022; see pg. 5).

3) In experiment 2 and 3, the main results are analyzed using a GLM approach. It is unclear why an ANOVA is not used here and how the main regressor was built. The latter may have implications for interpretation of the results.

We agree an ANOVA is appropriate here and thus report ANOVAs in Experiment 2 and 3 (pg. 12-23). The results remain the same.

Try to increase clarity in some points of the text. The use of the term discernment in the abstract is not clear as the term is only defined in the text.

We thank the reviewer for pointing this out. We now include a definition of discernment in the abstract (pg. 2).

State clearly in the Results section that experiment 2 and 3 are between subject (now it is only stated in the figure) along with the number of participants per group, including the number of excluded participants.

We now state that Experiment 2 and 3 are between-subject (pg.13, 21, 35-36). We also include the number of participants per group and excluded participants (pg. 7, 13-14, 21).

Include the interaction between type of response and valence in study 1. Report the results with and without political orientation to show that the results are independent of the inclusion of this factor.

Following the reviewer’s suggestion, we have repeated all the analyses without political orientation and include an interaction term for valence x type where valence is a factor. Once again, we find that discernment is greater in the ‘(Dis)Trust’ conditions than the other conditions (Supplementary files 1-12).

I would suggest that the DDM analysis either includes all model parameters as free or that the authors try to implement different variations (with different free parameters) and perform model comparison to test which is the most parsimonious fit to the data. In the recoverability analysis, I would report the correlation between the simulated and recovered parameters. And if different models are tested, I would include a confusion matrix of the recoverability of the models.

We now follow the reviewer’s recommendation and include all model parameters, including the starting point, as free. Once again, we find that the drift rate in the ‘(Dis)Trust’ environments (Experiment 2: v=0.216; Experiment 3: v=0.12) was meaningfully higher than the drift rate in both the ‘(Dis)Like’ (Experiment 2: v=0.01; 95% HDI of difference [0.048; 0.183], Experiment 3: 0.037; 95% HDI of difference [0.032;0.135]) and Baseline environments (Experiment 2: v=0.098; 95% HDI of difference [0.041; 0.195]; Experiment 3: v=0.006; 95% HDI of difference [0.061; 0.167]). By contrast there is no difference in drift rate between the latter two environments (Experiment 2: 95% HDI of difference: [-0.075; 0.08]; Experiment 3: 95% HDI of difference [-0.016; 0.079]). We do not find a difference in starting point bias across conditions in the main studies (Experiment 2 and 3) (see pg. 2328, Figure 6 and 7, Tables 1-2), though in the replications studies the baseline starting point is greater than in the ‘(Dis)Trust’ and/or ‘(Dis)Like’ conditions (see Supplementary files 15-18). In addition, as requested, we now report the correlations between real and recovered parameters (see Supplementary file 23).

When presenting the DDM results, I would avoid the systematic use of "good" and "bad" labels for the responses. It is fine to use this metaphor in one instance, but it sounds like an oversimplification when it is repeated many times.

As recommended by the reviewer we now changed the labels to “veracity promoting” and “veracity obstructing” (pg. 23-28 and pg. 41-42).

In the first paragraph of the discussion, the authors state "Users who found themselves in such an environment began sharing more true than false posts in order to receive carrots and avoid sticks." I would reformulate this statement not to imply learning. Although learning is implicit by the fact that the effects are a consequence of changing the incentive structure, the behavioral data analysis is aggregated across the whole experiment.

Following the reviewer’s suggestion, we now rephrased this sentence to *“Users who found themselves in such an environment, shared more true than false posts in order to receive carrots and avoid sticks*” (pg. 29)

Figure 7 is missing

We thank the reviewer for bringing this to our attention. Figure 7 was incorrectly labelled as Figure 8. We have now rectified this (pg. 27-28).

Reviewer #3:This paper suggests that changing the incentive structure on social platforms may influence people's discernment in their sharing decisions. Particularly, in experiment 1, the authors suggest that participants' use of trust-distrust buttons were more discerning than their use of like-dislike button. In experiments 2 and 3, the authors suggest that operating in a trust-distrust environment makes people's sharing more discerning than operating in a like-dislike or no-feedback environments. They also test a drift rate and claim that people place more weight to accuracy when provided with the distrust-trust feedback.Strengths:– The paper examines a novel mechanism to prevent misinformation spread on social platforms.– The authors' idea of offering buttons that are better linked to headline veracity is a practical solution that could be easily implemented.

We thank the reviewer for pointing out these strengths.

Weaknesses:– The conceptualization could be better articulated with wider coverage of the misinformation literature.

In the revised manuscript we better articulate the conceptualization (pg. 4-6) and cover more of the misinformation literature (pg. 5, 30-32).

– There is a disconnect between the theory and the empirical studies.

We find that the empirical findings follow our theory closely. In particular, our theory is that (i) people use ‘(Dis)Trust’ reaction buttons to discern true from false posts more so than the existing reaction buttons such as ‘(Dis)Like’. This is because trust by definition is the belief in reliability of information; and (ii) once users are exposed to an environment in which they can receive more ‘trusts’ carrots and fewer ‘distrust’ sticks for true than false posts they will share more true and less false information. Our empirical results show exactly that; (i) participants use ‘(Dis)Trust’ reaction buttons to discern true from false posts more than the existing ‘(Dis)Like’ reaction button options (Figure 3, pg. 10); and (ii) participants who are exposed to an environment in which they can receive more ‘trusts’ carrots and fewer ‘distrust’ sticks for true than false posts share more true and less false information than participants in other environments (Figure 5, pg. 18).

– Fake and true headlines may be different from one another on other factors (e.g., emotions) that affect people's trust judgments.

The reviewer raises the possibility that false and true headlines may differ systematically on factors such as emotion, and this difference may drive a trust judgement. We do not see this possibility as contradicting our conclusions. If people use a signal that is associated with veracity on social media platforms, such as emotion, to guide their discernment, that will still lead to a successful intervention in reducing misinformation spread. We note however, that there is no a-priori reason to assume that factors such as emotion will impact ‘(Dis)Trust’ reactions more so than ‘(Dis)Like’ reactions. Thus, we find this possibility less likely.

– The design of experiments 2 and 3 fails to accurately connect trust judgments to the veracity of the headlines.

‘Trust’ and ‘distrust’ judgments are directly connected to the veracity of the headlines. As described in the manuscript, participants use both ‘trust’ and ‘distrust’ buttons to discern between true and false posts (pg.9-10). Participants in Experiment 1 selected the ‘trust’ button more for true (M=18.897, SE=1.228) than false posts (M=9.037, SE=0.78; t(106)=9.846, p<0.001, Cohen’s d=0.952). They also selected the ‘distrust’ button more for false (M=24.953, SE=1.086) than true posts (M=9.252. SE=0.731; t(106)=16.019, p<0.001, Cohen’s d=1.549). These numbers of ‘trusts’ and distrusts’ per posts, which were gathered in Experiment 1, were then fed directly to the new groups of participants in Experiment 2 and Experiment 3 (see Figure 4, pg. 13, 36). Thus, they receive more ‘trusts’ for true than false posts, and more ‘distrusts’ for false than true posts.

Major CommentsConceptualization:Conceptually, authors suggest that adding buttons should make people's evaluations of headlines (in experiment 1) and the presence of much wider variety of feedback will make their sharing decisions better (in experiments 2 and 3).1. The reader would like to see better build-up of the theoretical framework. Some questions are:a. Whereas social media sharing (especially sharing 40-100 headlines) seem to be repetitive and can lead participants to develop habits (Anderson and Wood 2021; Ceylan et al., 2023). Do habits play a role? With the change in incentive structure, do participants develop different sharing habits?

Yes, changing the incentive structure could lead users to develop new sharing habits. We now note this possibility in the discussion (pg. 31).

b. Would we expect differential evaluation and sharing as a function of valence of the feedback button? The authors claim in many places that "… humans naturally seek social rewards and are motivated to avoid social punishments (Behrens, Hunt, Woolrich, and Rushworth, 2008; Bhanji and Delgado, 2014)." Do they test and find evidence for this? (More discussion and suggestions are below).

Yes, we expect and find that participants share differently based on the valence of the feedback button. Participants are expected to behave in a way that reflects a desire to attain more ‘trusts’ and ‘likes’ per post and fewer ‘distrust’ and ‘dislikes’ per posts. This is exactly what we find. In Experiment 1 more participants pressed ‘trust’/’like’ reaction buttons for true than false posts and more participants pressed ‘distrust’/‘dislike’ reaction buttons for false than true posts. In Experiment 2 participants were then fed the feedback which was gathered from (different) participants in Experiment 1. This means that participants in Experiment 2 would be able to maximize the average number of ‘trusts’/’likes’ they received per post by sharing more true posts and fewer false posts. They would also be able to minimize the average number of ‘distrusts’/’dislikes’ they received per post by sharing fewer false posts and more true posts. This is exactly what they do (Figure 5, pg. 18). This is true in Experiment 2 where only positively or only negatively valenced feedback is provided and also in Experiment 3 where both positively and negatively valenced feedback is provided together. The key is reducing the magnitude of negative feedback per post and increasing the magnitude of positive feedback per post. Just like in real life where a user does not simply want a ‘heart’ they want many ‘hearts’ (such as on Twitter) and they want the minimum number of ‘dislikes’ (such as on Youtube).

2. Figure 1b and 1c are rather confusing than clarifying. If I understand correctly, in experiment 2, participants receive one type of feedback that is disconnected from the actual veracity of the information. If people receive just distrust feedback for all the headlines (true and false), how is this in line with the idea that authors align carrots with truth and sticks with falsity?

Following the reviewer’s comment, it became apparent that the design and rationale behind Experiment 2 (pg.12-20) was unclear. The feedback participants receive is not disconnected from the actual veracity of the information. Let’s take the ‘distrust’ condition as an example. For any given post, true or false, some users will distrust the post. However, true posts receive fewer ‘distrusts’ than false posts. It is the number of ‘distrusts’ per post that matters. The participants in Experiment 2 (we assume) are motivated to minimize the average number of ‘distrusts’ they receive. To achieve this, they should post more true posts and fewer false posts. The same rationale holds for the group of participants that only receive ‘trusts’. They will receive more ‘trusts’ for true than false posts. It is the magnitude of ‘trusts’ that is associated with veracity. Again, this motivates participants to post more true and fewer false posts in order to maximize the average number of ‘trusts’ per post. Of course, if participants were simply trying of maximizing the total number of ‘trusts’ in Experiment 2, they would just repost on every trial. Participants do not do that. This indicates that they are sensitive to the number of ‘trusts’ per posts not just to the total number over all posts. Any user of social media platforms could relate to this. When posting a tweet, for example, many people will be disappointed with only a handful of ‘hearts’. The user’s goal is to maximize positive feedback per post. Similarly, if participants were simply trying of minimize the total number of ‘distrusts’ in Experiment 2 they would just skip on every trial. Participants do not do that, presumably because humans find the act of sharing information rewarding (Tamir and Mitchell, 2012). Thus, participants are motivated to share, but to do so while minimizing ‘distrusts’ per post. We now add this clarification to the revised manuscript (pg. 16-17).

Empirics:My overall feedback is that authors should better connect their conceptualization to their empirics.3. Why is the participant the evaluator of the news headline in experiment 1 and then the sharer of the headline in experiments 2 and 3?

This is a misunderstanding. Participants are different in each experiment. The participants in Experiment 1 are not the same participants as in Experiment 2, which are not the same participants as in Experiment 3. This is now explicitly stated on pg. 6, 12, 21.

4. The reader wonders where the action comes from: is it from reduction in rejecting false information or increasing in endorsing true information? In the model authors shared in the Supplemental Appendix Table 2, can the authors run a mixed effect model on the % of headlines engaged (not skipped) as a function of type of reaction and valence of reaction? This will help authors test their theory more robustly.a. The same for Supplemental Appendix Tables 5 and 7. Authors can test their theory by including an interaction term for type of reaction and valence of that reaction.

Following the reviewer’s request, we now add the interaction between type of reaction and valence of reaction to the above Tables. Note that the analysis is now an ANOVA rather than a mixed model, due to the request of Reviewer 2 to uses ANOVAs. What was Supplemental Appendix Table 2 (Experiment 1) is now Supplementary file 2. As a reminder, this table examined if the likelihood of clicking the different buttons was different for the different type of button options overall. Thus, the interaction asks if the likelihood of clicking the different button types differs as a function of valence. The answer is no. As can be seen, the interaction is not significant: F(1,106)=0.19, p = 0.64. We then do the same for what was Supplemental Appendix Tables 5 (Experiment 2), which is now Supplementary file 6. As a reminder, this Table originally asked if the likelihood of sharing posts was different across the different type of conditions. Thus, the interaction asks if the likelihood of sharing posts in the different conditions differs as a function of valence. The answer is no. The interaction is not significant: Experiment 2: F(1,311)=0.199, p = 0.656. Together the results suggest that the likelihood of reacting (Experiment 1) and the likelihood of reposting (Experiment 2) is not different for the different conditions as a function of valence options. As for what was Supplemental Appendix Table 7 (Experiment 3) valence of feedback does not exist as participants receive positive and negative feedback at the same time (‘Trust and Distrust’ and ‘Like and Dislike’), thus there is no interaction to be had.

5. In experiment 1, how did "skip" reaction impact people's discernment? Were those who skipped more discerning than those who used like-dislike buttons?

To test for a relationship between skipping and discernment we correlated across participants the percentage of posts skipped with discernment. We found that participants who skipped more posts were *less* discerning (R=-0.414, p<0.001). We now report this on pg. 11.

6. In experiments 2 and 3, the authors provide participants with 100 headlines. Participants make a skip or repost decision. Then, participants receive one of the types of feedback depending on their condition. One problem with this design and what authors suggest in Figure 1 is that participants receive one type of feedback in their respective condition. For instance, in the distrust condition, participants receive distrust feedback no matter what the veracity of the headline is. This is problematic because in this model, people do not have an opportunity to learn and connect veracity with trust feedback. I suggest authors could collect new data by cleaning up this mapping. The reader wonders how the results would change if this mapping actually happens and participants receive trust feedback when they share accurate information and distrust feedback when they share false information.a. This may explain why the authors do not find a differential effect between trust and distrust conditions. Trust feedback that is disconnected from the actual veracity of the headline would not enhance people's discernment differentially when this feedback is negative or positive. It seems like participants were just operating in an environment knowing that the feedback will be about trust but no matter what they share the valence of the trust would not change.

There seem to be a few misunderstandings here (see also response to comment 2 above). First, in Experiment 3 participants receive two types of feedback simultaneously, not one (either ‘trust’ and ‘distrust’ or ‘like’ and ‘dislike’). Second, the feedback in both Experiment 2 and Experiment 3 *is dependent* on veracity. What matters is the number of ‘trusts’ or ‘distrusts’ per post. True posts receive more ‘trusts’ (M=18.897, SE=1.228) than false posts (M=9.037, SE=0.78; t(106)=9.846, p<0.001, Cohen’s d=0.952) and false posts (M=24.953, SE=1.086) receive more ‘distrusts’ than true posts (M=9.252. SE=0.731; t(106)=16.019, p<0.001, Cohen’s d=1.549). This is the signal participants are picking up and learning from. Thus, participants indeed have an opportunity to learn veracity from ‘trust’ (and from ‘distrust’) feedback. Both ‘trusts’ and ‘distrusts’ are associated with veracity: greater number of ‘trusts’→ more likely to be true; smaller number of ‘distrusts’→ more likely to be true. There is no reason to apriori hypothesize that one condition (‘trust’ or ‘distrust’) will be better at driving discernment than the other. We try to make this clearer in the revised manuscript (pg. 16-17).

7. Further, in the current execution, participants learn about the reward and use this learning in their sharing decision all at one stage. However, the execution of the study can be cleaner if the authors separate the learning stage from the test stage (see Study 4 in Ceylan et al., 2023).

This is not entirely correct. First, participants make a decision to repost or skip a specific post (Post 1). This decision is final. Only then, do they receive feedback regarding Post 1. After this, they make a decision to repost or skip a completely different post (Post 2) and so on and so forth. Thus, they cannot use the feedback about Post 1 to make a decision about Post 1, because that decision has already been made. However, from the feedback about Post 1, they can learn about the consequences of posting true vs false posts and they can use this understanding to direct their next choice (Post 2). Our task is designed to mirror real life in this aspect. On Twitter and elsewhere, users do not spend a week (or a day) only making sharing decisions and another week (or day) only observing if others liked and retweeted their post. In any case, it is unclear why separating the stages will be cleaner or how exactly it may impact the conclusions.

8. One concern is that the authors stripped the headlines from any factor that could contribute people's truth and trust judgments. For instance, source credibility or consensus are factors that aid people's judgments on social platforms such as Facebook and Twitter.

Now that we have shown how an incentive structure can in principle reduce misinformation spread, it would be useful for future studies to test this intervention in real-world platforms and/or simulated platforms that include additional factors observed in real platforms. We make this point in the discussion (pg. 31).

One may wonder how people make their truth judgments in the absence of these cues. Can people discern true information from false information?

Yes, our results show that participants discern true from false information in the absence of cues such as the source of the information (see Figure 5, pg. 18 and Supplementary 5 – —figure supplement 1). The Y axis is discernment, which reflects the ability to discern true from false information. The (**) indicates that – yes – participants significantly discern true form false information in all studies when receiving ‘(Dis)Trust’ and ‘(Dis)Like’ feedback. In many cases we see they do so also when receiving no feedback. The thing to look out for is that the bars are significantly different from zero (Figure 5c, pg. 18). Crucially, they do so most when receiving ‘(Dis)Trust’ feedback.

In experiment 2, authors measure the discernment, but I did not see the means between true and false headlines.

We now report the means for true and false posts in Supplementary file 4.

Also, do people's trust and like responses correlate with the actual veracity of the information?

Yes, people trust and like true posts more than false posts. This can be observed by the corresponding bars measuring discernment in Experiment 1 (Figure 3, pg. 10). The fact that these bars are significantly different than zero indicated they like and trust true more than false posts. We also report this information in text on pg. 9 in the manuscript. Veracity is binary (true/false) rather than a continuum. Thus, the analysis is a t-test rather than a correlation.

9. Trust judgments also bear on other factors. One such factor is emotions (Dunn and Schweitzer, 2005). The reader wonders what the valence of the emotions is between the true and false headlines. A pretest that shows differences in emotions between false and true headlines would be helpful to the reader.

Whether and how emotion plays a role is an intriguing question. We hope this paper will prompt future studies to measure and test the role of emotion in the success of this intervention.

10. I am not very familiar with the computational analysis calculating drift rate. For readers who will not be very familiar with this method will need more detail about the analysis. Also, it seems like the results from experiments 2 and 3 do not show similar trends. Specifically, in experiment 2, the distance between the thresholds is the highest in the trust-distrust conditions and the non-decision time is the lowest. In experiment 3, there is no difference in distance between the thresholds across all three conditions. This time, the non-decision time was the highest in the trust-distrust conditions. How do the authors reconcile these differences?

In all sharing experiments (Experiment 2, Experiment 3, Experiment 5, Experiment 6), we observe a clear, significant and consistent difference in drift rate across conditions. This suggests that the rate of evidence accumulation toward ‘good judgments’ relative to ‘bad judgments’ is higher in the ‘(Dis)Trust’ environments than the other environments. This difference is consistently observed across all experiments and is thus the conclusion one should take away from the DDM analysis. As for other effects which are either observed as non-significant trends or are not replicable – we recommend caution in trying to interpret them. The reason we test the effects over and over, using slight differentiations across experiments, is that we wish to identify the most robust effects.